# Evidence of two differentially regulated elongasomes in *Salmonella*

Sónia Castanheira [1] & Francisco García-del Portillo [1✉]

Cell shape is genetically inherited by all forms of life. Some unicellular microbes increase niche adaptation altering shape whereas most show invariant morphology. A universal system of peptidoglycan synthases guided by cytoskeletal scaffolds defines bacterial shape. In rod-shaped bacteria, this system consists of two supramolecular complexes, the elongasome and divisome, which insert cell wall material along major and minor axes. Microbes with invariant shape are thought to use a single morphogenetic system irrespective of the occupied niche. Here, we provide evidence for two elongasomes that generate (rod) shape in the same bacterium. This phenomenon was unveiled in *Salmonella*, a pathogen that switches between extra- and intracellular lifestyles. The two elongasomes can be purified independently, respond to different environmental cues, and are directed by distinct peptidoglycan synthases: the canonical PBP2 and the pathogen-specific homologue $PBP2_{SAL}$. The PBP2-elongasome responds to neutral pH whereas that directed by $PBP2_{SAL}$ assembles in acidic conditions. Moreover, the $PBP2_{SAL}$-elongasome moves at a lower speed. Besides *Salmonella*, other human, animal, and plant pathogens encode alternative PBPs with predicted morphogenetic functions. Therefore, contrasting the view of morphological plasticity facilitating niche adaptation, some pathogens may have acquired alternative systems to preserve their shape in the host.

[1] Laboratory of Intracellular Bacterial Pathogens, National Centre for Biotechnology (CNB)-CSIC, Darwin 3, 28049 Madrid, Spain.
✉email: fgportillo@cnb.csic.es

Microbes have a defined shape inherited in their offspring. Bacteria exhibit a myriad of morphologies, including appendaged, helical, vibroid, branched, filaments, rod or coccoid shapes[1]. The defined shape inherited from generation to generation reflects genetic programmes that evolved to favour particular lifestyles[2–4]. A similar shape, however, does not imply phylogenetic relatedness. Bacteria of close phylogenetic groups can exhibit dissimilar shapes[1]. Conversely, *Escherichia coli* and *Bacillus subtilis*, belonging to the distant Proteobacteria and Firmicutes phyla, are both rod-shaped. Cellular asymmetry involving rod shape has been extensively studied in *E. coli* and *B. subtilis*, in which two multiprotein complexes, the elongasome and divisome, synthetise in a directional manner the peptidoglycan (PG) required for cell elongation and cytokinesis, respectively[5–8]. The elongasome and divisome share a common architecture: (i) an essential class b monofunctional PG synthase (bPBP) with transpeptidase activity; (ii) a member of the SEDS (shape, elongation, division and sporulation) protein family with glycosyltransferase activity[9]; and, (iii) adaptor proteins that connect the bPBP-SEDS module to a cytoskeletal scaffold.

In *E. coli*, PBP2 (MrdA) and RodA (MrdB) form the bPBP/SEDS module that inserts the new PG material required for cell elongation[9–12]. PBP2 and RodA are connected to short filaments of the actin homologue MreB that rotate along the cylindrical part of the cell[11,13]. The connection to a cytoskeletal scaffold ensures the growth of the PG sacculus with a defined topology. MreB interacts with the inner membrane proteins RodZ[14,15] and MreC/MreD[16]. The PBP2-RodA interaction is proposed to be modulated by MreC/MreD[17]. The bifunctional class a PG synthase (aPBP) PBP1a, with transpeptidase and glycosyltransferase activities, has also been implicated in repairing PG damage and promoting PG synthesis by the elongasome complex[18].

Besides PBP2, the intracellular bacterial pathogen *Salmonella enterica* serovar Typhimurium (*S.* Typhimurium) encodes a homologue named $PBP2_{SAL}$. This alternative bPBP shows a 63% identity to PBP2 uniformly spread along the sequence and conserves all motifs required for the transpeptidase reaction[19]. The in vivo data collected in a mouse infection model showed that the loss of $PBP2_{SAL}$ in *S.* Typhimurium leads to a slight increase in bacterial loads in target organs like the spleen and liver[19]. These data inferred a role of this bPBP in the adaptation to the host environment. Here, we examined $PBP2_{SAL}$ for a putative role in morphogenesis by analysing the shape of isogenic *S.* Typhimurium strains lacking PBP2 or $PBP2_{SAL}$ in different growth conditions. Our data support the existence in this pathogen of two elongasomes that respond to distinct environmental cues and are directed by each of these monofunctional bPBPs.

## Results

**PBP2 is dispensable in *Salmonella* in acidic pH.** *S.* Typhimurium has two monofunctional bPBPs that promote cell division independently, PBP3 and its homologue $PBP3_{SAL}$[20,21]. Like $PBP3_{SAL}$, the expression of $PBP2_{SAL}$ is upregulated in acidic pH[19,22] and both bPBPs show low affinity for beta-lactams when expressed ectopically at neutral pH[19]. $PBP2_{SAL}$ and $PBP3_{SAL}$ are produced by *S.* Typhimurium in vivo during the infection of mouse organs at relative levels that much exceed those of PBP2 and PBP3[19]. Within the host, PBP2 and PBP3 are indeed undetectable by Western blot[19,21]. Based on these observations, we sought to analyse whether $PBP2_{SAL}$ played a role in morphogenesis and its functional relationship with PBP2.

In *E. coli*, inhibition of PBP2 with beta-lactam antibiotics results in giant rounded cells unable to divide[23,24], which explains the essentiality of its coding gene, *mrdA*. We reasoned that if

$PBP2_{SAL}$ contributes to cell elongation in acidic pH, PBP2 could be dispensable under this condition. We successfully knocked-out *mrdA* in *S.* Typhimurium using acidified (pH 4.6) media. *mrdA* was deleted from nucleotide 51 to 1832 of the 1902 nucleotides encompassing the coding sequence (Supplementary Fig. 1a). Lack of PBP2 in Δ*mrdA* cells was confirmed by Western blot (Supplementary Fig. 1b). Whole-genome sequencing ruled out indels or missense/nonsense mutations in morphogenetic genes that could have suppressed lethality associated to the Δ*mrdA* deletion (Supplementary Table 1).

Once this *S.* Typhimurium Δ*mrdA* mutant was available, we examined morphological parameters in isogenic strains lacking PBP2 or $PBP2_{SAL}$ under growth conditions differing in the amounts of nutrients or the pH, neutral (7.0–7.4) or acidic (4.6). The loss of PBP2 or $PBP2_{SAL}$ did not affect the growth rate compared to wild-type bacteria except for Δ*mrdA* cells growing in LB pH 7.0, condition in which the culture reached lower final OD values (Supplementary Fig. 2). Consistently with the requirement of acidic pH for $PBP2_{SAL}$ production[19,22], Δ*mrdA* cells appear at neutral pH as giant spherical cells with larger size when growing in nutrient-rich LB medium in comparison to cells growing in the nutrient-poor PCN medium (Fig. 1a, b). These gross morphological alterations translated in decreased viability, ~5-log in LB and ~2-3 log in PCN media (Fig. 1a, b). The loss of rod shape in these growth conditions agreed with control assays showing the inability of $PBP2_{SAL}$ to promote cell elongation when expressed ectopically at neutral pH (Supplementary Fig. 3a). $PBP2_{SAL}$ activity therefore depends absolutely on acidic pH, a condition in which a mutation in its conserved serine S326 results in loss of function (Supplementary Fig. 3b). Unexpectedly, we detected by Western blot low levels of $PBP2_{SAL}$ and $PBP3_{SAL}$ in the spherical Δ*mrdA* cells growing at neutral pH (Fig. 1c). Such response could be triggered by the absence of all elongasome activity. Given the lack of activity of $PBP2_{SAL}$ and $PBP3_{SAL}$ at neutral pH[19], the fortuitous production of these alternative bPBPs may represent an alarm mechanism triggered to restore rod shape.

Unlike in neutral pH, the lack of PBP2 does not compromise cultivability in LB pH 4.6 medium, although some cells show defects in polar regions appearing as blunted edges, a phenotype more prominent in Δ*mrdA* cells than in wild-type or Δ$PBP2_{SAL}$ cells (Fig. 1a). In this growth condition, LB pH 4.6, Δ*mrdA* cells produce both PBP3 and $PBP3_{SAL}$ (Fig. 1c). This situation, implying one bPBP acting in cell elongation and two bPBPs promoting cell division, may alter the coordination between the elongasome and divisome complexes and result in aberrant morphologies at the midcell region. Remarkably, growth in minimal PCN pH 4.6 medium yielded Δ*mrdA* cells exhibiting a genuine rod shape with convex polar caps (Fig. 1b). This condition reproduces at a large extent the PBP2/PBP3 to $PBP2_{SAL}$/$PBP3_{SAL}$ switch that *S.* Typhimurium experiences in vivo in response to host cues[19]. Thus, bacteria growing in PCN pH 4.6 produce essentially $PBP2_{SAL}$/$PBP3_{SAL}$ instead of PBP2/PBP3 (Fig. 1c)[19]. The amount detected for these two latter enzymes, especially PBP2, is negligible in this growth condition (Fig. 1c). We also noted that Δ$PBP2_{SAL}$ cells show morphological alterations in PCN pH 4.6, with some cells appearing as rods but most exhibiting lemon-shaped forms (Fig. 1b). Δ$PBP2_{SAL}$ cells are indeed shorter compared to wild-type cells in the PCN pH 4.6 medium (Fig. 1d, e). The maintenance of some rod shape features in Δ$PBP2_{SAL}$ cells suggested that, although undetected by Western blot (Fig. 1c), a residual amount of PBP2 might be present to allow partial extension of the PG along the major axis. This hypothetical residual amount of PBP2 may also explain why wild-type cells are longer and thinner than Δ*mrdA* cells when growing in PCN pH 4.6 (Fig. 1b, d, e). Of note, whereas both

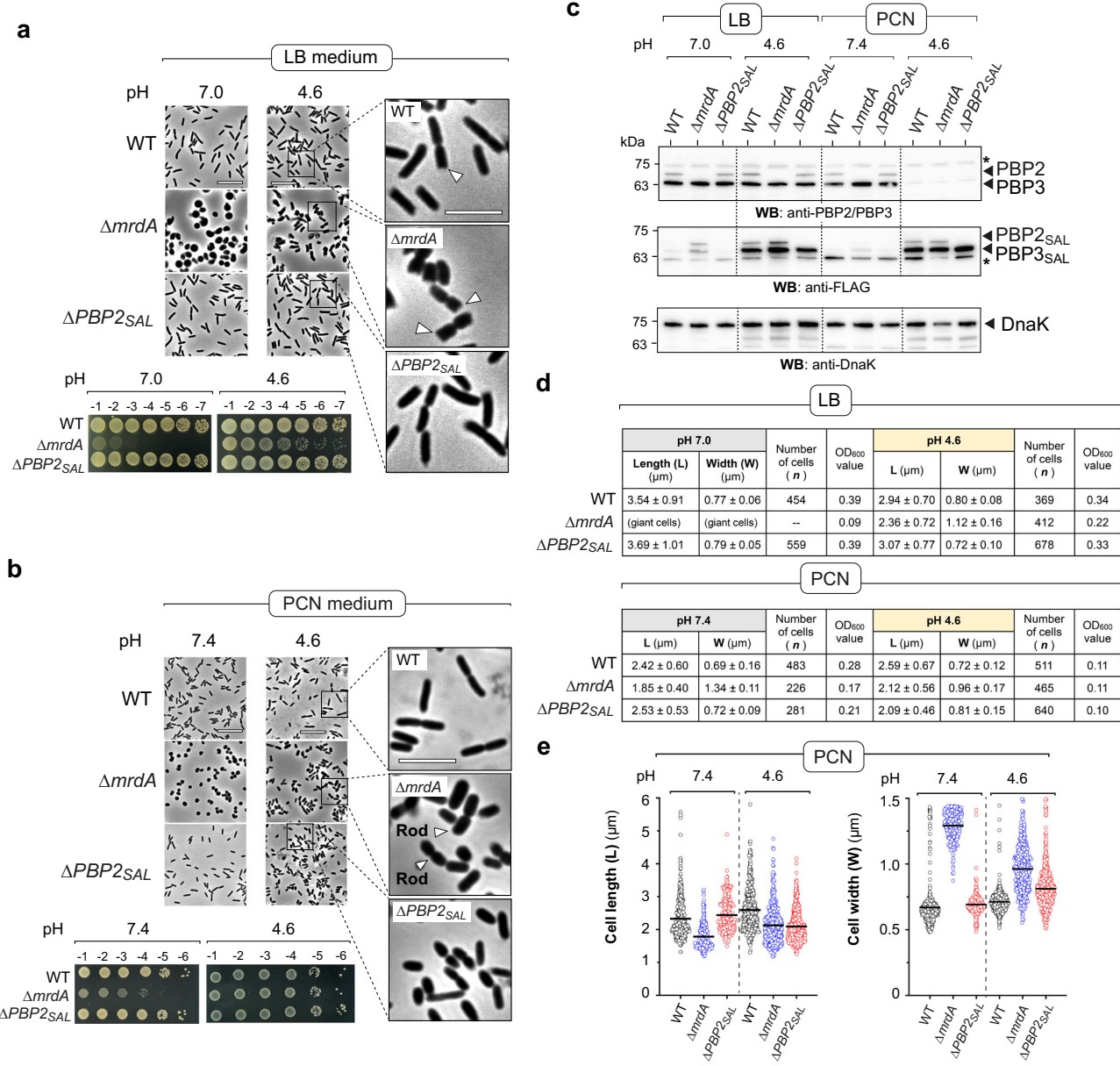

**Fig. 1 PBP2$_{SAL}$ generates rod shape in *Salmonella* independently of PBP2 in acidic pH. a** Morphology and viability of isogenic wild-type (WT), Δ*mrdA* and Δ*PBP2$_{SAL}$* *S*. Typhimurium strains grown in nutrient-rich (LB) media at neutral and acidic pH. Viability was assessed by spotting on LB plates serial dilutions (1:10) of the culture and overnight incubation at 37 °C (*n* = 3 biological). Magnifications are shown for the images taken in acidic pH. Arrowheads in the Δ*mrdA* cells point to polar caps with blunted edge. Scale bar: 10 μm. **b** Same as (**a**) but for bacteria grown in nutrient-poor PCN medium in neutral and acidic pH. Δ*mrdA* cells with normal rod shape are highlighted with arrowheads. Scale bar: 10 μm. **c** Levels of PBP2, PBP3, PBP2$_{SAL}$ and PBP3$_{SAL}$ in the indicated strains determined by Western blot in bacteria grown in LB and PCN media at neutral and acidic pH. These isogenic strains are 3xFLAG tagged at the 3'-end for the genes encoding PBP2$_{SAL}$ and PBP3$_{SAL}$ in their respective native chromosomal locations. Only exception is the Δ*PBP2$_{SAL}$* strain, which bears a single 3xFLAG tag in the PBP3$_{SAL}$-encoding gene. DnaK was detected as loading control. Samples correspond to total extracts of bacteria grown to the exponential phase. Asterisks (*) mark unspecific proteins detected with the antibodies. The positions and sizes of the molecular weight markers (in kDa) are indicated. Data are from a representative experiment (*n* = 2, biological). **d** Morphological parameters (average values of length and width) determined using ObjectJ (https://sils.fnwi.uva.nl/bcb/objectj/) for WT, Δ*mrdA* and Δ*PBP2$_{SAL}$* cells grown in the indicated media and pH. Included are details about the number of bacteria measured for each strain and condition and the OD$_{600}$ value at which cells were collected. Data are from a representative experiment (*n* = 2 biological). **e** Scatter dot plots showing the differences in length and width of WT, Δ*mrdA* and Δ*PBP2$_{SAL}$* cells grown in nutrient-poor PCN medium at neutral and acidic pH. Note the decrease in length in acidic pH associated to the loss of PBP2 or PBP2$_{SAL}$ (*n* = 2 biological).

Δ*mrdA* and Δ*PBP2$_{SAL}$* cells are shorter and wider than wild-type bacteria in PCN pH 4.6, only Δ*mrdA* cells exhibit a genuine rod shape with a clearly noticeable cylindrical region (Fig. 1b). Taken together, these data provided supporting evidence for PBP2$_{SAL}$ acting as morphogenetic protein that can generate rod shape independently of PBP2.

**PBP2$_{SAL}$ generates rod shape in *Salmonella* located inside eukaryotic cells.** *S*. Typhimurium produces PBP2$_{SAL}$ and PBP3$_{SAL}$ de novo when colonising the intracellular niche of host cells like macrophages[19,21] and fibroblasts (Supplementary Fig. 4). To dissect the role of PBP2$_{SAL}$ in this intracellular environment, we infected fibroblasts with the isogenic morphogenetic mutants previously

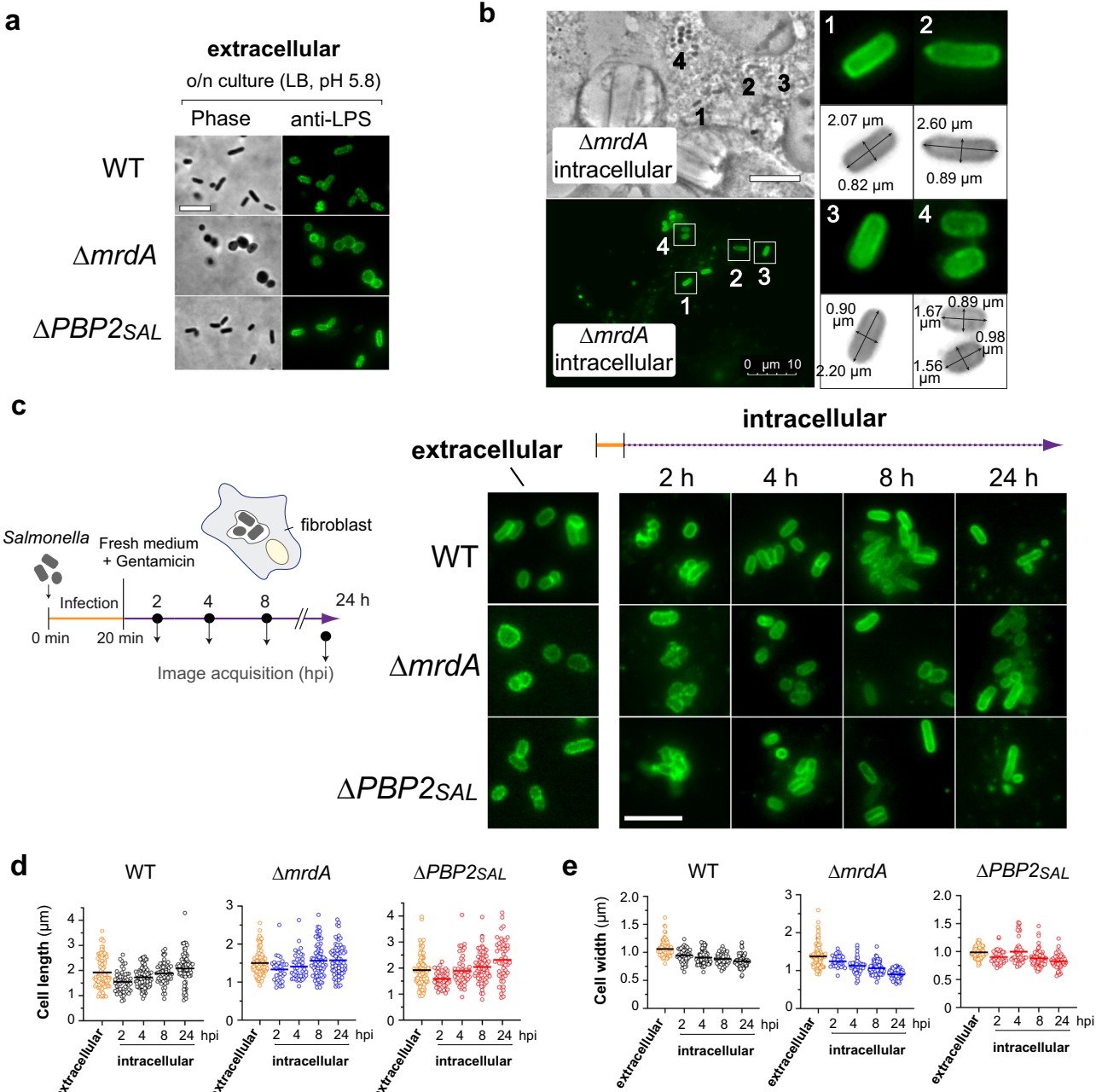

**Fig. 2 PBP2$_{SAL}$ responds to host signals to generate rod shape in intracellular *Salmonella*. a** Morphology of isogenic wild-type (WT), Δ*mrdA* and Δ*PBP2$_{SAL}$* cells in the input cultures after overnight culture in LB nutrient medium at pH 5.8. Scale bar, 5 μm. **b** Regain of rod shape by the Δ*mrdA* mutant at 8 h post infection of NRK-49F rat fibroblasts. Shown are the phase contrast image and the fluorescence obtained after labelling with anti-*S.* Typhimurium primary anti-O-atigen LPS rabbit antibody and a secondary anti-rabbit IgG antibody conjugated to Alexa488. Cellular dimensions (major, minor axes) of four representative rod-shaped intracellular bacteria (numbered from 1 to 4), are shown. (*n* = 2 biological). **c** Morphology of WT, Δ*mrdA* and Δ*PBP2$_{SAL}$* cells along the intracellular infection of NRK-49F rat fibroblasts. Shown is the fluorescence obtained after antibody-mediated labelling of the LPS. The readjustment in cell shape is noted in the Δ*mrdA* mutant from the early times post infection (2 h) (*n* = 2 biological). Scale bar, 5 μm. **d**, **e** Cellular dimensions (length, width) measured in the populations of input (extracellular bacteria) and intracellular bacteria at the indicated post infection times (2, 4, 8 and 24 hpi). Number of bacteria (*n*) measured in the distinct populations were for WT: 100 (extracellular), 54, 90, 58, and 83 (2, 4, 8 and 24 hpi); for Δ*mrdA*: 117 (extracellular), 32, 44, 87, and 71 (2, 4, 8 and 24 hpi); and, for Δ*PBP2$_{SAL}$*: 92 (extracellular), 44, 52, 82, and 51 (2, 4, 8 and 24 hpi). Major (length) and minor (width) axes were measured using ObjectJ (https://sils.fnwi.uva.nl/bcb/objectj/) and fluorescent images obtained after labelling with anti-LPS antibodies. The mean value in each sample is indicated by a thick horizontal line (*n* = 2 biological).

grown in LB medium at mild-acidic pH (5.8). At this pH, Δ*mrdA* cells do not exhibit normal rod shape but can divide (Fig. 2a). After 8 h of infection, some intracellular Δ*mrdA* cells showed a genuine rod shape with major versus minor axis ratios ≥2 and defined convex caps in the polar regions of the cell (Fig. 2b). Shape readjustment in these intracellular bacteria lacking PBP2 progressed

along the infection, being noticeable the rod shape from 8 h post infection and indistinguishable from the shape exhibited by wild-type cells or those lacking PBP2$_{SAL}$ (Fig. 2c). The average length and width measured in populations of intracellular bacteria showed that Δ*mrdA* cells regain rod shape morphology by decreasing width and with a minor net increase in length (Fig. 2d, e). Conversely,

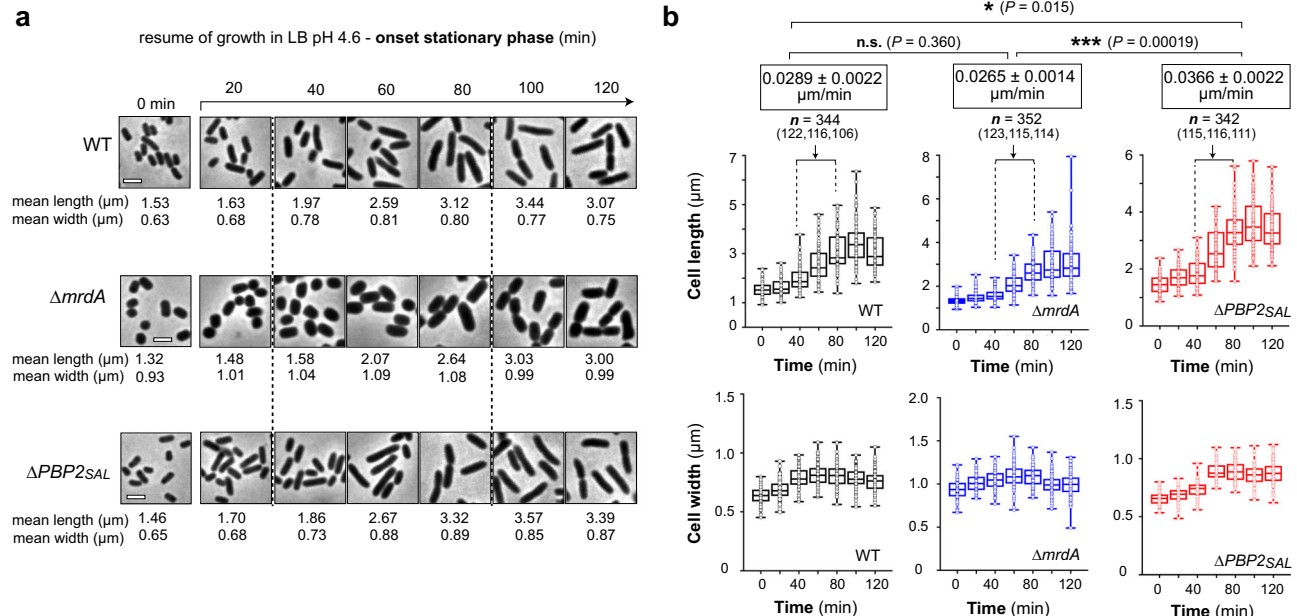

**Fig. 3 PBP2_SAL promotes cell elongation at a lower speed compared to PBP2 after resuming growth from the stationary phase. a** Mean length and width of wild-type (WT), $\Delta mrdA$ and $\Delta PBP2_{SAL}$ cells during the onset of stationary phase. Bacteria were grown overnight in LB pH 4.6, diluted in fresh medium to an initial $OD_{600} = 0.05$ and fixed at the indicated times for microscopy analysis. Number of cells measured ($n =$) at 0, 20, 40, 60, 80, 100 and 120 min were for WT: 111, 121, 122, 116, 106, 118 and 105 cells; for $\Delta mrdA$: 113, 113, 123, 115, 114, 119 and 132 cells; and for $\Delta PBP2_{SAL}$: 116, 110, 115, 116, 111, 107 and 107 cells ($n = 3$, biological). Scale bar: 2 µm. **b** Box-and-whisker plot representation for the indicated strains and times of mean length and width values with their standard deviation, this latter defined as the variation respect the mean of all individual cells measured for each morphological parameter (length or width).. The slope (length versus time ratio) was calculated by linear regression in the 40–80 min interval corresponding to the highest and constant growth rate ($n = 3$ biological). The significance of differences among slope values of the indicated strains was analysed using Free Statistics Calculator, version 4.0 (https://www.danielsoper.com/statcalc/calculator.aspx?id=103) (***$P < 0.001$; *$P < 0.05$; n.s. not significant).

intracellular wild-type cells increase their length with minor changes in width (Fig. 2d, e). In the case of $\Delta PBP2_{SAL}$ cells, the morphological alteration was basically the increase in length with no significant changes in width (Fig. 2d, e). Overall, these data demonstrated that PBP2_SAL can generate rod shape independently of PBP2 and inferred differences in their mode of cell elongation when these PG synthases act in the intracellular environment.

**PBP2_SAL and PBP2 elongate the cell at different velocities.** Given the apparent lower capacity of PBP2_SAL to elongate the cell during the residence of *S.* Typhimurium within host cell vacuoles (Fig. 2d), we were interested in determining the velocity at which PBP2 and PBP2_SAL insert new material into the PG. Average length and width values were measured in bacterial populations during growth at the onset of stationary phase, a condition in which cells undergo a quasi-synchronised first cell cycle. Cell elongation rates in the 40–80-min period (window time with constant growth rate) were estimated in 0.0289, 0.0265 and 0.0366 µm/min for wild-type, $\Delta mrdA$ and $\Delta PBP2_{SAL}$ cells, respectively (Fig. 3a, b). The statistical analysis, which included calculation of slopes from linear regression lines in the 40–80 min period ($n = 344$, $n = 352$ and $n = 342$ cells measured for wild-type, $\Delta mrdA$ and $\Delta PBP2_{SAL}$ strains), showed that the velocity at which PBP2 inserts PG material is significantly higher than that of PBP2_SAL (Fig. 3b).

The differences in PG elongation rates found between PBP2 and PBP2_SAL were confirmed by an alternative assay involving incorporation of the fluorescent molecule 3-[[(7-hydroxy-2-oxo-2*H*-1-benzopyran-3-yl)carbonyl]amino]-D-alanine hydrochloride (HADA)[25] in the PG of wild-type, $\Delta mrdA$ or $\Delta PBP2_{SAL}$ cells. HADA distribution was visualised after 1 min or 30 min uptake and at two pH values (7.0 and 4.6). HADA signal was mainly detected at the midcell region after a short uptake of 1 min, irrespective of the

genetic background or pH used (Fig. 4a, b). This result confirmed the midcell as the main region incorporating newly synthesised PG material in *S.* Typhimurium, as it was previously shown in *E. coli* with optimised labelling protocols that prevent removal of the HADA reagent by PG hydrolases[26]. With longer uptake times, 30 min, and pH 4.6 -a condition in which rod shape is attained by all strains used (see Fig. 1a, b)—the HADA signal remained stronger at the midcell region of $\Delta mrdA$ cells, confirming that the newly inserted PG is distributed along the cylindrical region of the cell by PBP2_SAL at lower rates compared to PBP2 (Fig. 4b, c). Wild-type bacteria showed a faint HADA signal at the midcell region whereas cells lacking PBP2_SAL exhibited a dispersed HADA labelling, indicative of PBP2 mediating rapid movement of newly inserted PG from the midcell (Fig. 4c). Analysis of HADA distribution with ObjectJ (https://sils.fnwi.uva.nl/bcb/objectj/) corroborated the accumulation of HADA label at the midcell region in wild-type cells, an effect more pronounced in the absence of PBP2 (Fig. 4d, e). Altogether, these data showed that PBP2 and PBP2_SAL are monofunctional bPBPs that elongate the PG at distinct rates.

**PBP2_SAL uses the MreBCD scaffold to insert PG material for cell elongation.** MreBCD are essential proteins in *E. coli* for cell elongation and are also required for growth in nutrient-rich media, although dispensable in nutrient-poor media[27]. Inactivation of *mreBCD* in *S.* Typhimurium resulted in cell rounding and, similarly to *E. coli*, affected the capacity to divide and to grow but only in nutrient-rich medium (Supplementary Fig. 5a, b). The lack of MreBCD components in cells producing exclusively PBP2_SAL ($\Delta mrdA$) led to the concomitant loss of rod shape in PCN pH 4.6 medium (Supplementary Fig. 6a, b), contrasting with the capacity of parental $\Delta mrdA$ cells to generate rod shape under this growth condition (Fig. 1b). A similar effect was observed for $\Delta PBP2_{SAL}$

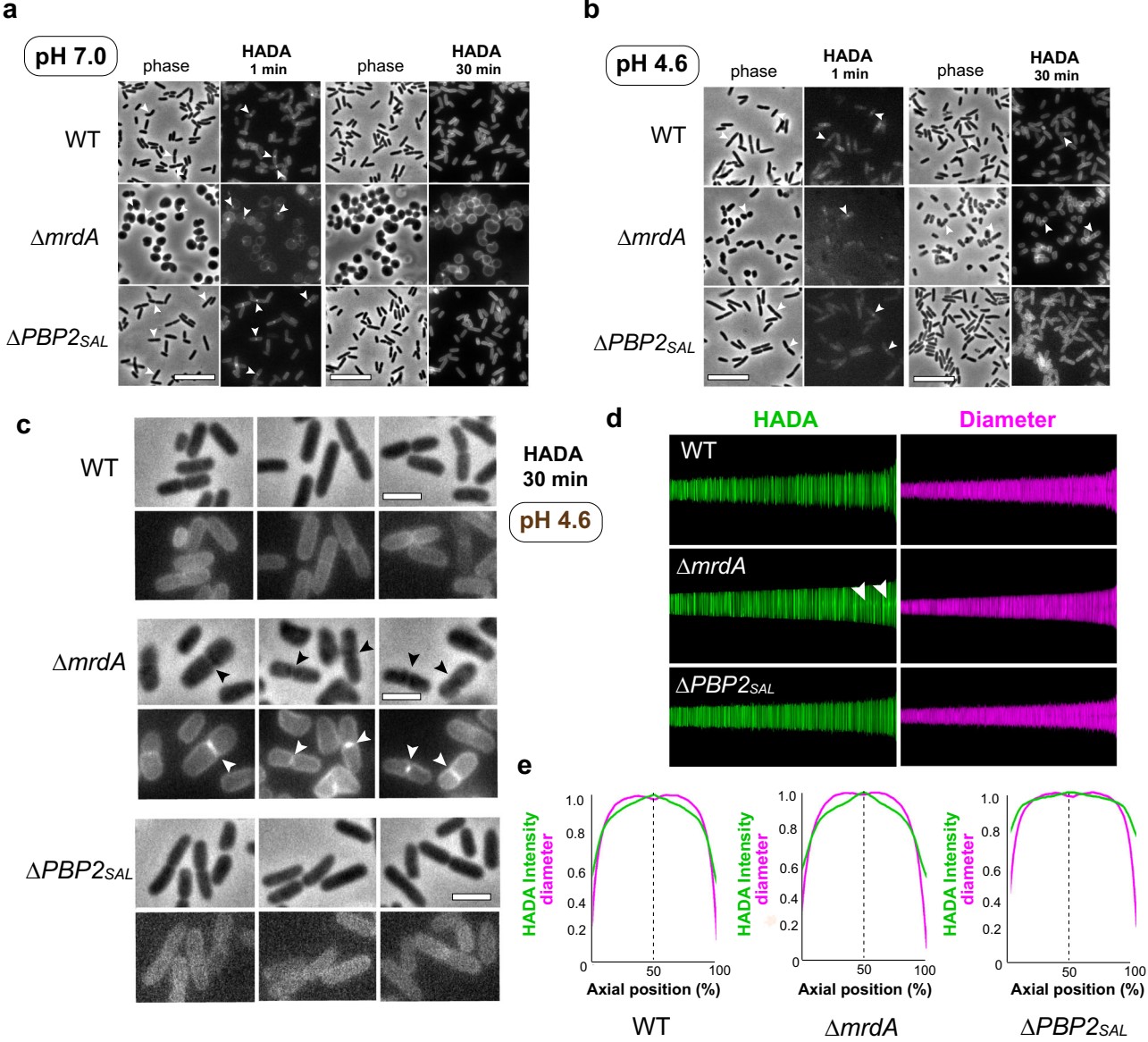

**Fig. 4 A different pattern of HADA incorporation supports changes in cell elongation rates between PBP2 and PBP2$_{SAL}$. a** Incorporation of 3-[[(7-Hydroxy-2-oxo-2$H$-1-benzopyran-3-yl)carbonyl]amino]-$_D$-alanine hydrochloride (HADA) in wild-type (WT), $\Delta mrdA$ and $\Delta PBP2_{SAL}$ cells during short or long times (1 min, 30 min, respectively). Bacteria were grown overnight at 37 °C in LB pH 5.8, diluted 1:50 in LB medium pH 7.0, grow for an additional 2.5 h, centrifuged and incubated for 1 or 30 min in fresh medium containing 125 μM HADA. At these times, bacteria were processed for immunofluorescence microscopy ($n = 2$ biological). Arrowheads indicate midcell sites in which a stronger HADA signal is visible. Scale bar, 10 μm. **b** Same as (**a**) but growing the strains in LB pH 4.6 ($n = 2$ biological). Scale bar, 10 μm. **c** Enlargement of the cells shown in (**b**) showing HADA incorporated for 30 min in WT, $\Delta mrdA$ and $\Delta PBP2_{SAL}$ cells grown in acidic pH. Scale bar, 2 μm. **d** Demographic map representation of the HADA fluorescence intensity (green) and width (magenta) of wild-type, $\Delta mrdA$ and $\Delta PBP2_{SAL}$ cells respect the medial axis. Arrowheads indicate increased fluorescence signal in the medial axis region of $\Delta mrdA$ cells. **e** Collective normalised profiles showing the distribution of HADA-derived fluorescence (green) and diameter/width (magenta) versus relative position along the cell axis.

cells lacking the MreBCD system (Supplementary Fig. 6c, d). Interestingly, both PBP2 and PBP2$_{SAL}$ are required for viability in the absence of the MreBCD system when bacteria grow in acidified minimal PCN medium (Supplementary Fig. 6a, c). Altogether, these data demonstrated that, in acidic pH, the generation of rod shape linked to the insertion of new PG material mediated by either PBP2 or PBP2$_{SAL}$ is supported by the MreBCD cytoskeletal scaffold.

**The PBP2- and PBP2$_{SAL}$-elongasomes assemble independently.**
The rod shape exhibited by $\Delta mrdA$ or $\Delta PBP2_{SAL}$ cells in acidic pH (Fig. 1a, b) suggested that PBP2 and PBP2$_{SAL}$ could elongate PG

independently. We hypothesised that these two PG synthases could be present in distinct elongasomes. To test this, we generated a strain expressing a functional PBP2$_{SAL-3xFLAG}$ tagged protein from its native chromosomal location to perform in vivo protein cross-linking with the outer membrane permeable cross-linker reagent disuccinimidyl suberate (DSS). Following subcellular fractionation, elongasome complexes were pulled down using anti-PBP2 or anti-FLAG antibodies from wild-type, $\Delta mrdA$ or $\Delta PBP2_{SAL}$ cells grown to mid-exponential phase in LB pH 4.6 medium. These immuno-precipitation assays showed that PBP2 and PBP2$_{SAL-3xFLAG}$ are components of distinct elongasomes. Thus, PBP2 is not pulled down with the anti-FLAG antibody (Fig. 5a, b) and, vice versa,

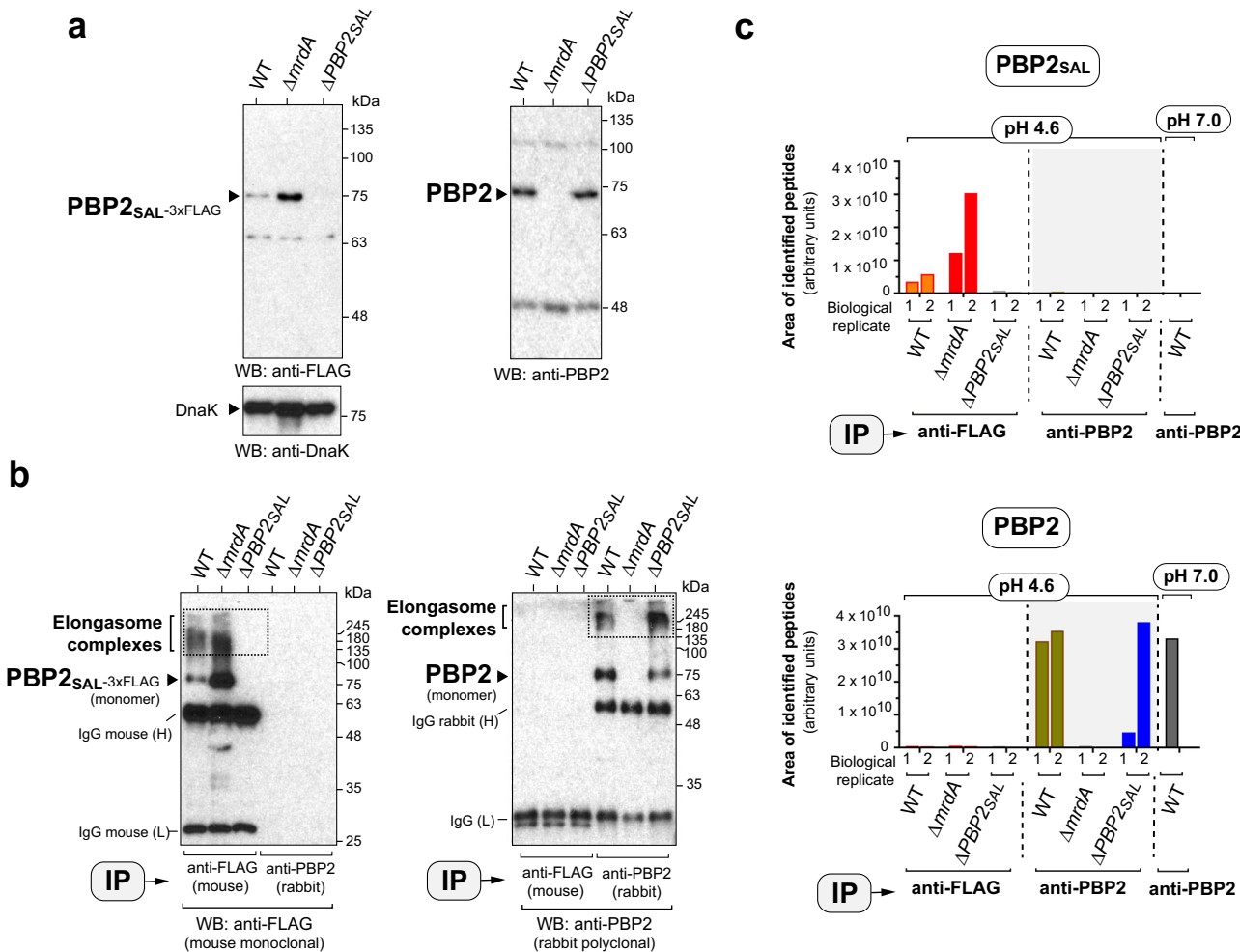

**Fig. 5 PBP2$_{SAL}$ and PBP2 assemble in independent elongasomes. a** Control Western assays showing the specificity of the anti-FLAG and anti-PBP2 antibodies used to pull down the PBP2$_{SAL}$- and PBP2 elongasomes. Shown are total protein fractions of the indicated strains grown to exponential phase (OD$_{600}$ ~0.5) in LB pH 4.6. The unrelated protein DnaK was detected in this input material as loading control. Positions and sizes of the molecular weight markers (in kDa) are indicated. **b** Pulldown of the PBP2$_{SAL}$- and PBP2-elongasomes from membrane fractions of the indicated strains. Wild-type (WT) and $\Delta mrdA$ cells bear an 3xFLAG-epitope tag at the 3′-end of the gene encoding PBP2$_{SAL}$ in its native chromosomal position. Bacteria were grown in LB pH 4.6, washed, and incubated for 30 min at room temperature (RT) in PBS pH 8.0 buffer containing the cross-linker reagent DSS (disuccinimidyl suberate). The cross-linking reaction was quenched with 20 mM Tris for 15 min at RT. Note the existence of independent elongasomes. In one case, the presence of PBP2$_{SAL-3xFLAG}$ in high-molecular complexes cross-linked with DSS in WT and $\Delta mrdA$ cells. In the other case, high-molecular weight complexes pulled down with anti-PBP2 antibody are only visible in WT and $\Delta PBP2_{SAL}$ cells ($n = 2$ biological). Bands corresponding to the heavy (H) and light (L) of the primary monoclonal mouse antibody anti-FLAG are indicated. IP, antibody used in the immunoprecipitation. Positions of the elongasome complexes and sizes of the molecular weight markers (in kDa) are also indicated. **c** Sum of the integrated areas of the peaks corresponding to the set of peptides identified for PBP2$_{SAL}$ or PBP2 (see "Supplementary Data" file) in the complexes immunoprecipitated (IP) from membrane fractions of bacteria grown at the indicated pH with anti-FLAG (PBP2$_{SAL}$) or anti-PBP2 antibodies ($n = 2$ biological). In the neutral pH condition, the pull down was performed exclusively with anti-PBP2 antibody given the no production of PBP2$_{SAL}$ by WT bacteria in this condition (see Fig. 1c). The proteomic data were processed using the Skyline v22.2 software (https://skyline.ms/project/home/software/Skyline/begin.view).

PBP2$_{SAL-3xFLAG}$ is not detected in the material pulled down with the anti-PBP2 antibody (Fig. 5a, b). The availability of $\Delta mrdA$ and $\Delta PBP2_{SAL}$ mutants was therefore instrumental to unequivocally demonstrate the existence of high-molecular elongasome complexes containing one or the other PG synthase.

The presence in *S.* Typhimurium of two elongasomes was further confirmed by proteomics of the high-molecular complexes pulled down from wild-type, $\Delta mrdA$ and $\Delta PBP2_{SAL}$ cells grown in acidic pH. Protein complexes analysed by proteomics were enriched in PBP2$_{SAL}$ and PBP2 peptides only in the respective positive controls: PBP2$_{SAL}$ peptides in samples pulled down with anti-FLAG antibodies from wild-type and $\Delta mrdA$ cells and, PBP2 peptides in the elongasome pulled down with anti-

PBP2 antibody in wild-type and $\Delta PBP2_{SAL}$ cells (Fig. 5c). Importantly, proteins reported to be elongasome components like RodA (MrdB), MreB, MreC, and RodZ were identified in the PBP2- and PBP2$_{SAL}$-elongasomes (Supplementary Fig. 7). MreB was fairly detected with a very low signal in the negative controls. MreB is an abundant protein that assembles into filaments that could have been pulled down minimally in an unspecific manner during the centrifugation steps. Indeed, MreB levels have been estimated in *E. coli* in ~2400–11,300 molecules per cell depending on the growth condition[28] and in *B. subtilis* in ~8000 molecules[29] whereas PBP2, at least in *E. coli*, represents one of the less abundant proteins and it is produced in the order of ~100 molecules per cell[30]. These differences may have affected the pull-

down assays considering that were performed, to the best of our knowledge for the first time, with all elongasome components produced at the physiological levels. On the other hand, we noticed that MreD, not identified in the samples, is a small protein of 162 aa residues that, when digested with trypsin, releases only one peptide (VLAMSIIAYLVALK) in the 8–20 aa size range normally analysed in standard proteomic searches. This feature may have hampered MreD detection by the standard trypsin-based mass spectrometry analyses. The presence in the immunoprecipitated complexes of the other cytoskeletal proteins (MreB, MreC) as well as RodA and RodZ (Supplementary Fig. 7), demonstrated that, although physically independent, the PBP2- and PBP2$_{SAL}$-elongasomes share the basic components needed for cell elongation.

**A fraction of the PBP2 and PBP2$_{SAL}$ elongasomes contain the bifunctional PG synthase PBP1b and D-alanyl-D-alanine carboxypeptidases (D,D-CPases).** Rod shape with a defined length-to-width ratio was proposed in *B. subtilis* and *E.coli* to result from the balance between oriented PG synthesis by the bPBP-RodA system and unoriented insertion of new PG material by unconnected bifunctional aPBPs[31]. Disbalance of these activities leads to more elongated or wider cells[31]. The loss in *S.* Typhimurium of PBP2 or PBP2$_{SAL}$ in cells growing in acidic pH results in changes in length and width (Figs. 1–4), which supported altered ratios between oriented and unoriented PG synthesis. The proteomic data obtained from the PBP2- and PBP2$_{SAL}$-elongasomes showed the presence of the bifunctional PG synthase PBP1b in both complexes (Supplementary Fig. 8), inferring a connection between the oriented and unoriented modes of PG growth. Intriguingly, PBP1b was barely detected in the PBP2-elongasome assembled by wild-type cells at neutral pH (Supplementary Fig. 8). We also searched for the presence of PBP1a, the other major bifunctional aPBP proposed to contribute to cell elongation in *E. coli*[18]. However, PBP1a peptides were identified with much lower abundance (ca. ~tenfold less intensity in the overall signals, see Supplementary Fig. 8 and "Supplementary Data" file). Furthermore, PBP1a was better detected in the PBP2$_{SAL}$-elongasome (Supplementary Fig. 8). Based on these findings, we constructed a series of isogenic strains combining deficiencies in PBP2 or PBP2$_{SAL}$ with those in PBP1a or PBP1b. The morphological parameters of this collection of isogenic strains in media with different amount of nutrients (LB, PCN) and at neutral and acidic pH did not show significant differences associated to the loss of PBP1a or PBP1b (Supplementary Figs. 9 and 10). Indeed, these single and double mutants cultured in media differing in amount of nutrients or pH did not show marked growth defects in any condition (Supplementary Figs. 11 and 12). The fact that the loss of either aPBP (PBP1a or PBP1b) did not translate in major changes in shape indicated that both aPBPs can contribute to insertion of new PG material in coordination with any of the two elongasome complexes.

We also examined the proteomic data for identifying low-molecular weight PBPs with D-alanyl-D-alanine carboxypeptidase (D,D-CPase) and endopeptidase activities: DacA (PBP5), DacB (PBP4), DacC (PBP6), DacD (PBP6b) and PbpG (PBP7)[32]. These enzymes were identified in the elongasomes pulled down from wild-type, Δ*mrdA* and Δ*PBP2$_{SAL}$* cells with enrichment of DacB and DacC in the PBP2$_{SAL}$-elongasome, i.e., identified with better signal-to-noise scores in the Δ*mrdA* cells respect the other strains (Supplementary Fig. 8 and "Supplementary Data" file). Among this group of enzymes, the endopeptidase PbpG (PBP7) was identified with the lowest signal and only with a few peptides (Supplementary Fig. 8), making unlikely its association to the elongasome. Interestingly, in *E. coli* the D,D-CPase activity of

## Table 1 Bacterial strains used in this study.

| Strain/plasmid | Relevant genotype (*) | Source/reference |
|---|---|---|
| *S.* Typhimurium | | |
| SV5015 | SL1344 *his*G$^+$ | 63 |
| MD5052 | Δ*mrdA* (Δ*PBP2*) | 19 |
| MD2576 | Δ*PBP2$_{SAL}$* | 19 |
| MD5054 | *PBP2$_{SAL}$*-3xFLAG | This work |
| MD2259 | *PBP3$_{SAL}$*-3xFLAG::Km$^R$ | 21 |
| MD5510 | Δ*mrdA PBP2$_{SAL}$*-3xFLAG | This work |
| MD5549 | Δ*mrdA* pAC-6xHIS-*PBP2$_{SAL}$* | This work |
| MD5550 | Δ*mrdA* pAC-6xHIS-*PBP2$_{SAL}$*$^{S326A}$ | This work |
| MD5057 | Δ*mrdA* pFUS | This work |
| MD5058 | Δ*mrdA* pFUS-*PBP2* | This work |
| MD5059 | Δ*mrdA* pFUS-*PBP2$_{SAL}$* | This work |
| MD6296 | *PBP2$_{SAL}$*-3xHA::Cm$^R$ | This work |
| MD6297 | *PBP2$_{SAL}$*-3xHA::Cm$^R$ *dacA*-3xFLAG::Km$^R$ | This work |
| MD6298 | *PBP2$_{SAL}$*-3xHA::Cm$^R$ *dacB*-3xFLAG::Km$^R$ | This work |
| MD6299 | *PBP2$_{SAL}$*-3xHA::Cm$^R$ *dacC*-3xFLAG::Km$^R$ | This work |
| MD6300 | *PBP2$_{SAL}$*-3xHA::Cm$^R$ *dacD*-3xFLAG::Km$^R$ | This work |
| MD6505 | *PBP2$_{SAL}$*-3xHA::Cm$^R$ *fepA*-3xFLAG::Km$^R$ | This work |
| MD5577 | Δ*mreB*:: Km$^R$ | This work |
| MD5578 | Δ*mreC*:: Km$^R$ | This work |
| MD5579 | Δ*mreD*:: Km$^R$ | This work |
| MD5580 | Δ*mreCD*:: Km$^R$ | This work |
| MD5581 | Δ*mreBCD*:: Km$^R$ | This work |
| MD5583 | Δ*PBP2$_{SAL}$* Δ*mreB*:: Km$^R$ | This work |
| MD5584 | Δ*PBP2$_{SAL}$* Δ*mreBCD*:: Km$^R$ | This work |
| MD5585 | Δ*mrdA* Δ*mreB*:: Km$^R$ | This work |
| MD5588 | Δ*mrdA* Δ*mreBCD*:: Km$^R$ | This work |
| MD2591 | Δ*mrcA* (Δ*PBP1a*) | This work |
| MD2569 | Δ*mrcB* (Δ*PBP1b*) | This work |
| MD6264 | Δ*mrdA* Δ*PBP1a*:: Km$^R$ | This work |
| MD6274 | Δ*mrdA* Δ*PBP1b*:: Km$^R$ | This work |
| MD3201 | Δ*PBP2$_{SAL}$* Δ*PBP1a*:: Km$^R$ | This work |
| MD6283 | Δ*PBP2$_{SAL}$* Δ*PBP1b*:: Km$^R$ | This work |
| MD5064 | *PBP2$_{SAL}$*-3xFLAG *PBP3$_{SAL}$*-3xFLAG | 19 |
| MD5516 | Δ*mrdA PBP2$_{SAL}$*-3xFLAG *PBP3$_{SAL}$*-3xFLAG::Km$^R$ | This work |
| MD5098 | Δ*PBP2$_{SAL}$* *PBP3$_{SAL}$*-3xFLAG::Km$^R$ | This work |
| *E. coli* | | |
| BW25141 | Δ*lacZ4787*(::*rrnB*-3) Δ(*phoB-phoR*)580 λ⁻ *galU95* Δ*uidA3*::*pir*$^+$ *recA1 endA9*(del-ins)::FRT *rph-1* Δ(*rhaD-rhaB*)568 *hsdR514* | 66 |
| MD6291 | BW25141 pSU314::3xHA | This work |

*All *S.* Typhimurium strains listed are isogenic and derivates of parental strain SV5015.

DacB (PBP4) decreases in acidic pH[33], which makes probable that its connection to the *S.* Typhimurium elongasomes could be more associated to this accessory endopeptidase activity. In the case of DacD (PBP6b), its presence in the PBP2- and PBP2$_{SAL}$-elongasomes (Supplementary Fig. 8) is consistent with the reported preferential use of this enzyme by *E. coli* in acidic pH[33] and its upregulation by intracellular *S.* Typhimurium[34]. No other endopeptidases like MepM or MepS were detected in our proteomic analyses of the two elongasome complexes.

The unexpected identification of some D,D-CPases in the PBP2- and PBP2$_{SAL}$-elongasomes was further evaluated in a parallel assay. Isogenic strains producing simultaneously a 3xFLAG-tagged variant of the respective D,D-CPase and a PBP2$_{SAL-3xHA}$ variant, were constructed (Table 1). All tags were introduced in the native chromosomal locations of the respective genes. Elongasomes were pulled down with either anti-PBP2 or anti-HA(PBP2$_{SAL}$) antibodies to search for the presence of the 3xFLAG-tagged D,D-CPases. These assays, performed in wild-type cells, proved the presence of DacA, DacB, DacC and DacD in the PBP2- and

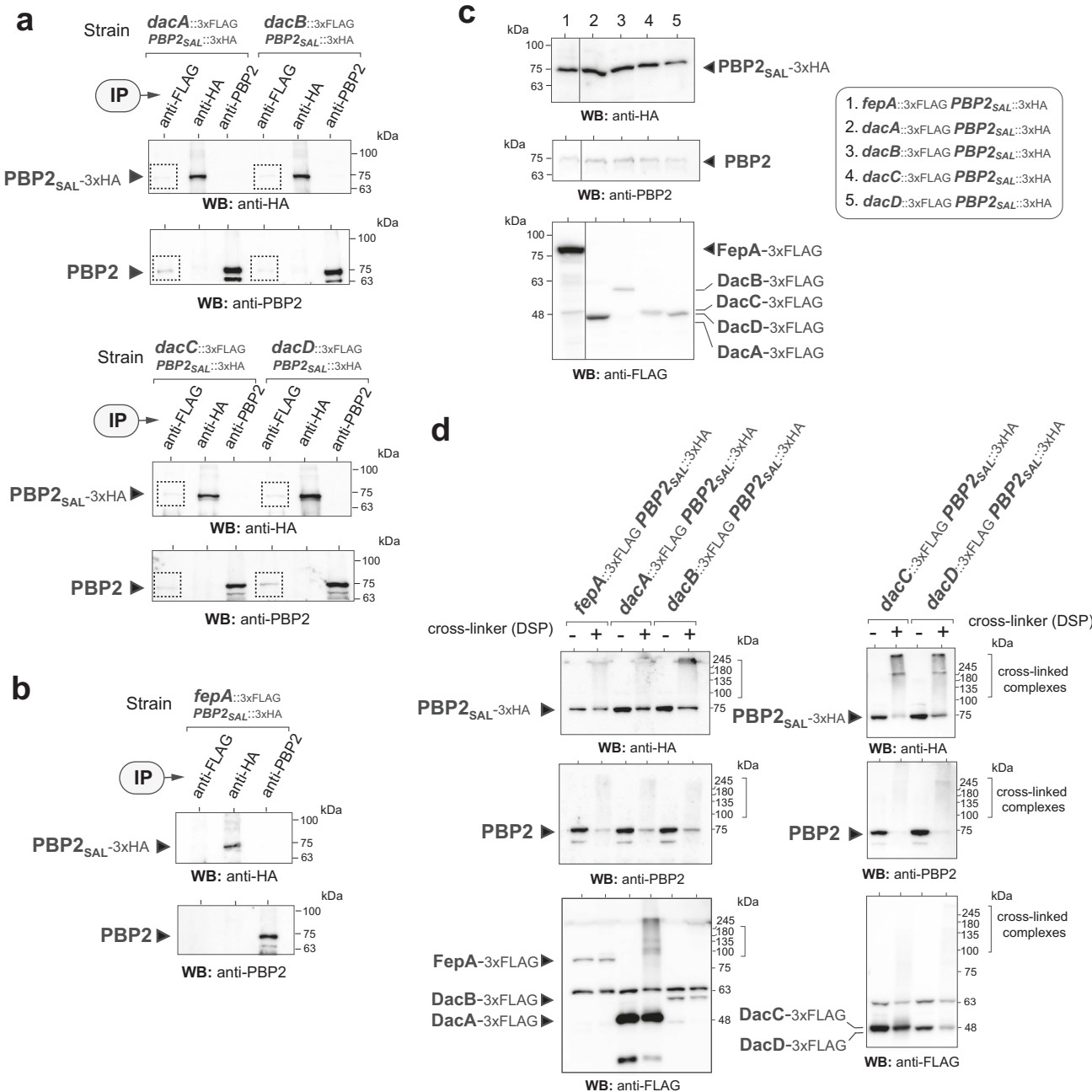

**Fig. 6 The D,D-CPases DacA, DacB, DacC and DacD are present in a fraction of PBP2- and PBP2$_{SAL}$-elongasomes. a** Immuno-precipitates of elongasome complexes from wild-type cells grown in LB pH 4.6. The isogenic strains used for these assays, all in wild-type genetic background and bearing a 3xHA epitope tag in the 3′ end of the PBP2$_{SAL}$-encoding gene, bear an 3xFLAG epitope in the respective genes encoding D,D-CPases (*dacA, dacB, dacC, dacD*) ($n = 2$ biological). Complexes were cross-linked in vivo with the reagent DSP (dithiobis[succinimidylpropionate]), immunoprecipitated (IP) with the indicated antibodies and the gels run in denaturing conditions, containing β-mercaptoethanol. The presence of PBP2$_{SAL}$ or PBP2 in complexes pulled down with the anti-FLAG antibodies recognising D,D-CPases is highlighted with boxes ($n = 2$ biological). IP, antibody used in the immunoprecipitation. Positions and sizes of the molecular weight standards (in kDa) are indicated. **b** Control immunoprecipitation assays in a strain bearing a 3xFLAG epitope in the unrelated inner membrane FepA, involved in iron transport ($n = 2$ biological). Positions and sizes of molecular weight standards (in kDa) are indicated. **c** Relative levels in the input samples of the proteins under study (PBP2$_{SAL}$, PBP2, DacA, DacB, DacC, DacD, FepA) in the membrane extracts prepared prior to the in vivo cross-linking ($n = 2$ biological). Two different areas of the blots were spliced together as indicated by the line shown between lines 1 and 2 of the blots. Positions and sizes of the molecular weight standards (in kDa) are indicated. **d** Presence of the proteins under analysis (PBP2$_{SAL}$, PBP2, DacA, DacB, DacC, DacD, FepA) in higher molecular weight complexes following cross-linking with DSP. Samples are compared in bacteria exposed or not to cross-linker and the gels run in the absence of β-mercaptoethanol ($n = 2$ biological). Positions and sizes of molecular weight standards (in kDa) are indicated.

PBP2$_{SAL}$-elongasomes (Fig. 6a). However, the PBP2 or PBP2$_{SAL}$ signals obtained in the Western blots were much lower when pulling down with the anti-FLAG (D,D-CPase) antibody than when immunoprecipitating with anti-HA(PBP2$_{SAL}$) or anti-PBP2

antibodies (Fig. 6a). We interpreted this difference as wild-type bacteria having a only a small fraction of the elongasomes with D,D-CPase(s) bound to them. This fraction could increase for the PBP2$_{SAL}$-elongasome, in which DacB and DacC were identified

with higher signals in the proteomic analyses ($\Delta mrdA$ cells, see Supplementary Fig. 8 and "Supplementary Data" file). As a negative control, we used a strain producing an unrelated inner membrane, the iron transporter FepA. The FepA-$_{3xFLAG}$-tagged variant was not pulled down with anti-PBP2 or anti-HA antibodies despite being produced at high levels (Fig. 6b, c). Additional cross-linking assays revealed that the four D,D-CPases identified in a small fraction of the elongasomes of wild-type cells (DacA, DacB, DacC and DacD) can be detected in high-molecular weight complexes (Fig. 6d). However, unlike PBP2 and PBP2$_{SAL}$, only a small fraction of the D,D-CPases is cross-linked in high-molecular complexes (Fig. 6d), a finding consistent with the low amount of PBP2 and PBP2$_{SAL}$ pulled down when using anti-FLAG(D,D-CPase) antibodies (Fig. 6a). Taken together, these data supported the presence of D,D-CPases and endopeptidases in a fraction of the PBP2- and PBP2$_{SAL}$-elongasomes, with higher relative abundance of DacB(PBP4) and DacC(PBP6) in the PBP2$_{SAL}$-elongasome.

## Discussion

Most bacteria show invariant shape during their life cycle, conservation assigned to a universal morphogenetic system with three interconnected modules[2,3,9,35]. These include a cytoskeletal scaffold formed by homologues of actin (MreB) or tubulin (FtsZ), PG synthases that insert material in a precise orientation and, adaptor proteins that tether the cytoskeletal guide to the PG biosynthetic module. Variations consisting in additional proteins or unique changes in the PG can result in cell curvature. That is the case of *Caulobacter crescentus* and *Helicobacter pylori* using additional cytoskeletal proteins[36–38] or, *Vibrio cholerae* that generates curvature by producing a periplasmic structure that directly binds to the cell wall[39]. In rod-shaped *E. coli* and *B. subtilis* cells, MreB and FtsZ polymers direct cell elongation and division, respectively[35,40,41]. PBP2a-RodA/PBP2b-FtsW are the biosynthetic modules in *B. subtilis* and PBP2-RodA/PBP3-FtsW in *E. coli*. An early study in *B. subtilis* reported an apparent redundant action of two bPBPs, PBP2a and PBPH, for generating rod shape[42]. The extensive characterisation of these model bacteria concluded in a widely accepted model in which rod shape is generated by one elongasome and one divisome that alternate in the insertion of PG material longitudinally or transversally, respectively.

Our findings challenge this dogma based on the presence in a bacterium of two elongasomes capable of generating rod shape independently and in response to distinct cues. This view contrasts the widely accepted link between morphological variation and increased adaptability[43], as it occurs in dimorphic fungi[43,44] and a few bacterial pathogens that alter shape upon contacting host cells[43]. Our cross-linking assays provide solid evidence for the presence in *S.* Typhimurium of two independently assembled elongasomes that share similar cytoskeleton and adaptor modules (MreB, MreC, RodZ) and the glycosyltransferase (RodA) required for incorporating new PG material. Besides the different bPBP directing the complex, the higher abundance of D,D-CPases and endopeptidases in the PBP2$_{SAL}$-elongasome could be another distinguishable feature that adjusts the rate at which the PG grows. Another potential distinct feature between the two elongasomes could be the PBP2-RodA and PBP2$_{SAL}$-RodA complexes, which may exhibit unique structural features since the optimal pH at which they function is different. A recent study in *E. coli* has shown that the PBP2-RodA complex is highly dynamic, transiting between open closed and open states and being this latter the one that promotes RodA glycosyltransferase activity[12]. This study, as well as a previous one performed with a PBP2-RodA complex of *Thermus thermophilus*[45], report in vitro polymerisation assays at neutral pH. Given the strict dependence

for the acidic pH of the *S.* Typhimurium PBP2$_{SAL}$, it will be worth to analyse in future studies the polymerisation activity of the PBP2-RodA and PBP2$_{SAL}$-RodA complexes at both neutral and acidic pH. This approach could provide clues on how the PG polymerisation activity of the complex is modulated depending on the bPBP involved and the pH.

The different lifestyles of *S.* Typhimurium as a free-living planktonic bacterium or as intracellular pathogen may have contributed to fix alternative morphogenetic programmes. Rod-shaped bacteria confined within the phagosomal compartment must necessarily coordinate PG elongation to the growth of the surrounding host membrane, a constraint not experienced by free-living bacteria. This assumption considers the distinct velocities measured for the PBP2- and PBP2$_{SAL}$-elongasomes. Maintenance of an integral phagosomal membrane may require a deceleration in the speed at which PG is elongated from early post-infection times upon entry into the host cell. Future studies are required to unravel how PG metabolism could be functionally linked to virulence functions that intracellular *S.* Typhimurium exploit to either rupture or preserve the phagosomal membrane[46–49].

With regards to the production of bPBPs, our Western blot data were insightful, confirming the PCN pH 4.6 medium as the most accurate to mimic the environment that *S.* Typhimurium faces in the host (Fig. 1c). It is remarkable the coincidence of much higher amounts of PBP2$_{SAL}$/PBP3$_{SAL}$ over PBP2/PBP3 in bacteria growing in acidified PCN medium (Fig. 1c) as it is observed in vivo in bacteria infecting the host[19]. This interchange of morphogenetic bPBPs occurring in vivo is consistent with the residence of *S.* Typhimurium within acidic phagosomes along the infection process[50].

A puzzling observation of our in vitro assays, which is not reflected in the in vivo situation[19], is however the production of the two morphogenetic systems (PBP2/PBP3 and PBP2$_{SAL}$/PBP3$_{SAL}$) in comparable amounts by wild-type cells growing in acidified LB medium (Figs. 1a, c and 5b). The bases of this phenomenon are currently unknown, and if not having available mutant cells lacking PBP2 or PBP2$_{SAL}$, it could certainly have hampered the identification of two elongasomes as different entities. We also repeatedly observed in PCN pH 4.6 that the cultures reached low-density values even after an overnight incubation (Supplementary Fig. 2). This behaviour resembles that exhibited by intracellular *S.* Typhimurium in vivo, in which it has been repeatedly reported the formation of slow-growing persister cells within phagosomal compartments[51,52]. Strikingly, the slow growth exhibited by *S.* Typhimurium in PCN pH 4.6 triggers the switch to the pathogen-specific monofunctional bPBPs (Fig. 1c) and slow growth has been reported to be a determinant factor that triggers persistence in this pathogen[53]. These findings support a connection between PBP2$_{SAL}$/PBP3$_{SAL}$ and the establishment of persistent infections in the intracellular niche, a tempting hypothesis that merits to be investigated.

Intracellularity has imposed changes compared to free-living bacteria in the mode that morphogenetic proteins interact. *Chlamydiae*, obligate intracellular pathogens naturally devoid of FtsZ, use MreB for cytokinesis[54]. *Chlamydiae* could use for division the only two PBPs encoded by its genome, PBP2 and PBP3, since both interact with MreB and the cell division protein FtsK[55]. Whether PBP2 and PBP3 act in the same or independent divisomes with MreB, is however unknown. Considering that the PBP2- and PBP2$_{SAL}$-elongasomes of *S.* Typhimurium use MreBCD, alternative morphogenetic complexes with a common cytoskeletal platform may certainly exist in other bacteria.

The identification of the PG bifunctional synthase PBP1b in the PBP2- and PBP2$_{SAL}$-elongasomes was unexpected given that in *E. coli* this aPBP interacts with the division proteins FtsN and PBP3[56]. Other studies have however indirectly linked PBP1b to the elongasome. Increased PBP1b levels reduce susceptibility to the

PBP2-targeting β-lactam mecillinam[57] and *E.coli* cells producing a thermosensitive PBP2 and lacking PBP1b rapidly lyse if exposed to mecillinam[58]. PBP1b is also required for recovery of rod shape in *E. coli* spheroplasts lacking PG after lysozyme treatment[59]. In *E. coli*, PBP1a and PBP1b were reported to be favoured in alkaline and acidic conditions, respectively[60]. However, the loss of PBP1a or PBP1b does not compromise *S.* Typhimurium viability or shape in acidic pH and both synthases seem to be interchangeable in the PBP2- and PBP2$_{SAL}$-elongasomes.

Our cross-linking assays show that D,D-CPases and endo-peptidases might be present in a fraction of PBP2- and PBP2$_{SAL}$-elongasomes. D,D-CPases and endopeptidases generate tetra-peptides that can participate as acceptors in the transpeptidation reaction catalysed by mono- and bifunctional PBPs. The balance between these reactions, transpeptidase of the aPBP and bPBP compared to D,D-CPase/endopeptidase activities, could affect the cell elongation rate. Interaction among enzymes having distinct activities in PG metabolism was recently shown for MltG, a lytic transglycosylase that interacts with aPBPs and for which short-ening of newly inserted nascent glycan chains was proposed[61]. In a recent Tn-seq study performed in *Acinetobacter baumannii*, DacC (PBP5) was also linked to the elongasome[62], although its putative function in the complex was declared as unclear.

We conclude that PBP2$_{SAL}$ may have facilitated *S.* Typhimurium adaptation to the intracellular lifestyle by reducing the cell elongation rate and allowing a better coordination with the rate at which the phagosomal membrane grows. This tentative model also considers a putative connection with other virulence-related functions given the increase in bacterial loads in target organs infected with *S.* Typhimurium lacking PBP2$_{SAL}$[19]. bPBPs homologues to PBP2$_{SAL}$ exist in other bacterial genera representing mostly animal and plant bacterial pathogens and, opportunistic pathogens (Supplementary Fig. 13). Among these are included the *Citrobacter, Klebsiella, Serratia, Edwardsiella, Pantoea, Hafnia, Erwinia, Kluyvera, Mixta, Ranhella, Raoultella* and *Shimwellia* genera. The presence of alternative morphogenetic systems in these genera is of utmost importance for the development of innovative antimicrobial therapies, considering that they are absent in most of the known beneficial microbiota.

## Methods

**Bacterial strains and plasmids used in the study**. The *S.* Typhimurium strains, all isogenic to wild-type strain SV5015[63], that were used in this study are shown in Table 1. The plasmids used are listed in Table 2.

**Bacterial growth conditions**. *S.* Typhimurium strains were grown in Luria-Bertani (LB) broth, composed of 1% (w/v) casein peptone, 0.5% (w/v) yeast extract and 0.5% (w/v) sodium chloride. The nutrient-poor phosphate-carbon-nitrogen (PCN) minimal medium[64], was also used. The composition of PCN medium is 4 mM Tricine [N-[Tris (hydroxymethyl) methyl]glycine],

**Table 2 Plasmids used in this study.**

| Plasmid | Relevant markers | Source/reference |
|---|---|---|
| pFUS | Km$^R$ | 74 |
| pAC-6xHIS-P$_{lac}$ | Cm$^R$ | 74 |
| pKD13 | Km$^R$, Amp$^R$ | 66 |
| pKD46 | Amp$^R$ | 66 |
| pSUB11 | 3xFLAG sequence, Km$^R$ | 67 |
| pCP20 | FLP$^+$, Amp$^R$, Cm$^R$ | 75 |
| pSU314 | Cm$^R$ | 67 |
| pMA-T::3HA-SacI | Amp$^R$ | This work |
| pSU314::3xHA | Cm$^R$ | This work |

0.1 mM FeCl$_3$, 376 µM K$_2$SO$_4$, 15 mM NH$_4$Cl, 1 mM MgSO$_4$, 10 µM CaCl$_2$, 0.4% (w/v) glucose, 0.4 mM K$_2$HPO$_4$/KH$_2$PO$_4$, and micronutrients. When required, pH was buffered with 80 mM MES [2-(N-morpholino) ethanesulfonic acid]. To grow strains bearing genetic elements conferring antibiotic-resistance, media were supplemented with chloramphenicol (10 µg/mL); kanamycin (30 µg/mL or 60 µg/mL, in neutral or acidic pH, respectively); or ampicillin (100 µg/mL). To induce the production of the iron transporter FepA, bacteria were grown in LB medium pH 4.6 containing 50 µM of the iron chelator diethylenetriamine-pentaacetic acid (DTPA), as described[65]. To monitor growth patterns, bacteria were incubated at 37 °C in the different media and pH by triplicate using 96-well plates, 120 µL volume of culture and an initial optical density value at 600 nm (OD$_{600}$) of ~0.01. Changes in OD$_{600}$ were automatically registered every 20 min with 20 s of orbital agitation using a Tecan Spark microplate reader. These experiments were repeated in a minimum of two biological replicates.

**Molecular genetics procedures**. The oligonucleotides used as primers in the PCR reactions are listed in Table 3. PCR reactions were performed using Q5 polymerase (New England Biolabs) according to manufacturer instructions. PCR fragments and gel extractions were purified using the NucleoSpin™ Gel and PCR Clean-up kit (Macherey-Nagel), and plasmid DNA was purified using the Macherey-Nagel™ NucleoSpin Plasmid QuickPure™ kit.

**Generation of *S.* Typhimurium isogenic mutants lacking morphogenetic proteins**

*ΔmrdA null mutant.* The *mrdA* clean deletion strain was generated by one-step λ-Red recombination[66] for insertion of a kanamycin resistance (Km$^R$) cassette, previously amplified from pKD13 template plasmid[66] using primers KO-STPPBP2-FW and KO-STPPBP2-RV (Table 3). The Km$^R$ cassette was subsequently removed with plasmid pCP20[66] yielding MD5052 (ΔmrdA) strain (Table 1).

*ΔmreB and ΔmreBCD mutants.* To delete *mreB* using the one-step λ-Red recombination procedure[66], a Km$^R$ cassette was amplified from pKD13 plasmid using primers KO-*mreB*-FW/KO-*mreB*-RV (Table 3). The primers KO-*mreB*-FW/KO-*mreD*-RV were used to delete *mreBCD* using the same procedure (Table 3). The PCR products were purified and electroporated into *S.* Typhimurium strains SV5015, ΔmrdA and ΔPBP2$_{SAL}$ strains (Table 1). Recombinants were selected on PCN pH 7.4 plates containing 30 µg/mL Km or, for the case of the ΔmrdA genetic background, PCN pH 5.8 plates containing 60 µg/mL Km.

*S. Typhimurium PBP2$_{SAL-3xHA}$-Cm-tagged strain.* Integration of the of the 3xHA tag at the 3'end of the gene encoding *PBP2$_{SAL}$* at its native chromosome location was carried out by the λ-Red recombination procedure modified for tagging purposes[66,67]. Synthetic DNA corresponding to the 3xHA-SacI tag sequence and received as plasmid pMA-T-3xHA-SacI (Invitrogen) was digested with SacI and cloned in vector pSU314[67] digested with SacI. The resulting plasmid pSU314::3xHA-Cm$^R$ was used as DNA template to amplify a GlySerGlySer-3xHA-Cm$^R$ cassette with PBP2$_{SAL}$-GlySerGlySer-3xHA-FW/PBP2SAL-3xHA-Cm-RV primers (Table 3).

*Doubly-tagged (3xFLAG, 3xHA) strains in genes encoding PBP2$_{SAL}$ and D,D-CPases.* Isogenic strains were constructed by chromosomal tagging using the λ-Red recombination procedure to insert 3xFLAG-Km$^R$ cassettes at the 3' end of genes *dacA, dacB, dacC* or *dacD* encoding D,D-carboxypeptidases that cleave

**Table 3 Oligonucleotides used in this study.**

| Name | Sequence (5′ → 3′) |
|---|---|
| KO STPBP2 FW | CAACCACCGGGTTTTCCGCAGGCAAATGGGTGTTGTTGTCGCCCAGCATGGTGTAGGCTGGAGCTGCTTC |
| KO STPBP2 RV | ATGAAACGACAAAATTCTTTTCGTGACTATACGGCTGAGTCCGCACTGTTATTCCGGGGATCCGTCGACC |
| fwSpeI-PBP2 | CGCTACTAGTATGAAACGACAAAATTCTTTTCGTG |
| revSpe-PBP2 | CCCTACTAGTTTATTGGTCCTCCGCCGCTGCAACC |
| fwSpeI-PBP2SAL | CGCTACTAGTATGACTTTTAAAGACTTTGATGCGG |
| revSpeI-PBP2SAL | AAGGGCTTTATCCTCCGGCGGCAACGGTTAAACCCTATAT |
| PBP2SAL S326A FW | ATATAGGGTTTAACCGTTGCCGCCGGAGGATAAAGCCCTT |
| PBP2SAL S326A RV | AAGTAAGCGGATTTTGTTTTCCGCCCCAGCTTTCAGGATTATCCCTTAGTGTGTAGGCTGGAGCTGCTTC |
| KO mreB FW | GATCATTTCCAGCGCCTTGCCGCCGCCGCGGCCACACAGGTCAGCGGAT ATTCCGGGGATCCGTCGACC |
| KO mreB RV | AAACTGCTGGCGTACTTTACGCATCAGCAAAAAGAGCCAGGGCCACAGCAATTCCGGGGATCCGTCGACC |
| KO MreD RV | ACAAAGGAAACGAACTGCACTAATT |
| mreB-FL FW | GGCCACGGCTAAAAATTGGCTTCAT |
| mreB-FL RV | GAGTCAATAATTCCTGGCGACGCGG |
| mreD-FL RV | GAGTCAATAATTCCTGGCGACGCGG |
| PBP2SAL-GlySerGLySer-3xHA FW | ACCACCTGTTTGATCCACAGGCTGATACCACACAGCCGGATCAGGCGCCAGGATCTGGATCTTACCCATATG |
| PBP2SAL-3xHA-Cm RV | ATGCAGATATTCCGGCCGTATCCTTGTCTGATGGCGCTTTGCTTATTTGATGTAGGCTGGAGCTGCTTCG |
| KO-PBP 1A Fw | GCGTCTGGCTTGCCTTTATACTACCGCGCGTTTGTTTATAAACTGCCCAAGTGTAGGCTGGAGCTGCTTC |
| KO-PBP 1A Rv | GCCGTCATCCGCTAAACACAATAAAAAAGGCGCCGGAGCGCCTTTTTTGAATTCCGGGGATCCGTCGACC |
| KO-PBP 1B Fw | GGAAGAATAGAGAATCGGGCCTTTGCGCCTGTATGTTGCGGAGAAAAAGCGTGTAGGCTGGAGCTGCTTC |
| KO-PBP 1B Rv | TAACGGTGTAGACCGGGTAAGCACAGCGCCACCCGGCACTATTACCGTGAATTCCGGGGATCCGTCGACC |
| FL-1A-Fw | AGCCATAACGGTCGATCTCC |
| FL-1A-Rv | AGGGCCGAATTGCCTGATGG |
| FL-1B-Fw | TCGTGACGTGTTATACGTTGCCTC |
| FL-1B-Rv | GACGTAAGCGTCTTATTCGGCCTA |

the terminal D-Ala-D-Ala bond in stem peptides of the pepti-doglycan. An additional 3xFLAG-tagged strain at the 3' end of the unrelated *fepA* gene, encoding an iron transporter, was also constructed as control. The doubly-tagged strains were generated by transducing with P22 H105/1 *int201* phage the 3xFLAG-Km$^R$-tagged alleles of the respective D,D-CPases-encoding genes and MD6296 (*PBP2$_{SAL}$*-3xHA-Cm$^R$) as recipient strain (Table 1).

*Doubly-tagged strains producing PBP2$_{SAL}$-3xFLAG and PBP3$_{SAL}$-3xFLAG variants.* A P22 H105/1 *int201* phage lysate obtained in strain MD2559 (*PBP3$_{SAL}$-3xFLAG*-Km$^R$)[21] was used to transduce the *PBP3$_{SAL}$-3xFLAG*-Km$^R$ allele to recipient strain MD5510 (Δ*mrdA PBP2$_{SAL}$-3xFLAG*), generating the doubly-tagged strain MD5516 (Δ*mrdA PBP2$_{SAL}$-3xFLAG PBP3$_{SAL}$-3xFLAG* -Km$^R$) (Table 1). An additional isogenic-related tagged strain was constructed using as recipient MD2576 (Δ*PBP2$_{SAL}$*) to generate MD5098 (Δ*PBP2$_{SAL}$ PBP3$_{SAL}$-3xFLAG*::Km$^R$) (Table 1). The doubly-tagged strain MD5064 (*PBP2$_{SAL}$-3xFLAG PBP3$_{SAL}$-3xFLAG*) in wild-type background was available from our previous study[19].

*PBP2$_{SAL}$ variant with point mutation in catalytic serine (S326).* The PBP2$_{SAL}$ variant bearing a point mutation in the catalytic serine (S326) was generated by overlap extension PCR using genomic DNA of wild-type *S.* Typhimurium SV5015 strain as a template. Fragments 1 and 2 were amplified using primers fwSpeI-PBP2SAL/PBP2SAL-S326A-RV for fragment 1 and PBP2SAL-S326A-FW/revSpeI-PBP2SAL for fragment 2 (Table 3). The resulting PCR product was purified, digested with SpeI and ligated into pAC-HIS plasmid previously digested with SpeI and treated with alkaline phosphatase enzyme (Roche) to avoid plasmid self-ligation. PBP2$_{SAL}$ with S326A mutation was verified by sequencing.

*Strains defective in PBP1a or PBP1b.* The *mrcA* and *mrcB* genes encoding PBP1a and PBP1b, respectively, were deleted by the λ-Red recombination procedure[66] using primers KO-PBP1A-Fw/KO-PBP1A-Rv and KO-PBP1B-Fw/KO-PBP1B-Rv (Table 3) in pKD13 template. The Km$^R$ cassette was removed with plasmid pCP20 to generate strains MD2591 (Δ*mrcA*) and MD2569 (Δ*mrcB*) (Table 1).

Deletions were confirmed by PCR and sequencing using primers FL-1A-Fw/FL-1A/Rv and FL-1B-Fw/FL-1B/Rv. The Δ*mrcA*::Km$^R$ and Δ*mrcB*::Km$^R$ alleles of the intermediate strains were transduced by P22 H105/1 *int201* phage to MD5052 (Δ*mrdA*) and MD2576 (Δ*PBP2$_{SAL}$*) strains to generate the respective double mutants listed in Table 1.

**Whole-genome sequencing of the Δ*mrdA* strain and SNP analysis.** Genomic DNA from *S.* Typhimurium strain MD5052 (Δ*mrdA*) grown in LB pH 4.6 medium was conducted by "Parque Científico de Madrid" (https://fpcm.es/). Paired-end sequencing (2 × 150) was performed using the Illumina Miseq platform (Illumina, Inc.). The number of reads was 1,638,199 with total number of bases of 491,658,478 (% GC = 52.24), representing a ×115 coverage for a genome of size 4.2 Mb. Alignment of raw sequences to the *S.* Typhimurium strain SL1344 reference genome (NCBI entry no. FQ312003.1) was performed with the Burrows-Wheeler aligner (https://bio-bwa.sourceforge.net/) with default parameters for paired-end reads. Samtools/bcftools[68] were used to compress, sort, index, and detect single nucleotide polymorphisms (SNPs) from alignment results in BAM format. The biological impact of the detected SNPs was determined by snpEff (https://pcingola.github.io/SnpEff/). The Integrative Genomics Viewer (IGV) browser (https://www.igv.org/) was used to visualise and to select relevant SNPs.

**Microscopy analysis of morphogenetic mutants.** At the desired growth conditions, bacteria were fixed with 3% (w/v) paraformaldehyde (PFA) for 10 min and adjusted to a final PFA concentration of 1%. These fixed bacteria were centrifuged (4300 × *g*, 2 min, RT) and resuspended in phosphate-buffered saline (PBS), pH 7.4. A volume of 50 µL of the bacterial suspension was dropped on poly L lysine-precoated coverslips and incubated for 15 min at RT. The attached bacteria were washed three times with PBS, and the coverslip was mounted on slides using ProLong Gold Antifade (Molecular Probes). Images were acquired on an inverted Leica DMI 6000B microscope with an automated CTR/7000 HS

controller (Leica Microsystems) and an Orca-R2 charge-coupled-device (CCD) camera (Hamamatsu Photonics).

**Infection of fibroblasts.** NRK-49F (ATCC CRL-1570) normal rat fibroblasts were propagated at 37 °C in a 5% $CO_2$ atmosphere in Dulbecco's modified Eagle medium (DMEM) supplemented with 10% foetal bovine serum (FBS). When reaching 70% confluence, the fibroblasts were infected in 24-well tissue culture plates with *S.* Typhimurium morphogenetic mutants, previously grown overnight in 1.5 ml of LB pH 5.8 in static conditions. These overnight bacteria were centrifuged at $8000 \times g$, and resuspended in Hank´s balanced salt solution (Gibco) to an $OD_{600}$ of 1.0. and used to infect the fibroblasts at a multiplicity of infection of 10:1 (bacteria:fibroblast). The 24-well plates were centrifuged for 5 min at $500 \times g$ at room temperature to ensure efficient invasion of all strains. Infected fibroblasts were incubated at 37 °C in a 5% $CO_2$ incubator for 20 min and then treated up to 2 h post infection with 100 µg/mL gentamicin to kill extracellular bacteria. Gentamicin concentration was reduced to 10 µg/mL thereafter.

**Morphological analyses at the onset of the stationary phase.** The *S.* Typhimurium morphogenetic mutants were grown overnight at 37 °C in 1.5 mL of LB pH 4.6, diluted to an $OD_{600}$ of 0.05 in 25 mL LB pH 4.6 and grown at 37 °C for ~120 min. Every 20 min after dilution, the $OD_{600}$ was measured and 1 mL of each cell culture was fixed with 3% PFA for microscopy analysis and determination of morphogenetic parameters using ObjectJ (https://sils.fnwi.uva.nl/bcb/objectj/)[69].

**Immunofluorescence microscopy.** NRK-49F fibroblasts were seeded on coverslips to a confluence of 50–60% in 24-well plates and infected with wild-type, Δ*mrdA* and Δ*PBP2_SAL* strains. Infected cells were fixed at 2, 4, 8 and 24 hpi in 3% PFA (15 min, RT), and processed for immunofluorescence microscopy, as described[70]. Briefly, after PFA fixation, the infected cells were washed in PBS pH 7.4 and incubated for 10 min at RT in blocking solution containing 0.1% (w/v) saponin and 1% (v/v) goat serum. Incubations with primary and secondary antibodies were performed in this same solution of 0.1% (w/v) saponin and 1% (v/v) goat serum during 45 min each, with three washes in PBS pH 7.4 after the respective incubations. Coverslips were finally mounted on slides using ProLong Gold Antifade (Molecular Probes). Rabbit polyclonal anti-*S.* Typhimurium LPS (cat. no. 229481, 1:1000, Difco Antiserum-BD Diagnostics, Sparks, MD) and goat polyclonal anti-rabbit IgG conjugated to Alexa 488 (1:1000, cat. no. A-11008, ThermoFisher Scientific), were used as primary secondary antibodies, respectively. Images were acquired on an inverted Leica DMI 6000B fluorescence microscope with an automated CTR/7000 HS controller (Leica Microsystems) and an Orca-R2 charge-coupled-device (CCD) camera (Hamamatsu Photonics).

**Drop-spotting viability assay.** *S.* Typhimurium morphogenetic mutants were inoculated in an appropriate volume of minimal PCN pH 5.8 medium and the cultures grown at 37 °C overnight with aeration (150 rpm shaking conditions). Overnight cultures were adjusted to similar $OD_{600}$ values to subsequently prepare serial 1:10 dilutions in 0.85% (w/v) NaCl. Five microlitres of each dilution were spotted onto agar plates with either nutrient-poor PCN or nutrient-rich LB media. Plates were incubated overnight at 37 °C.

**HADA incorporation to monitor the distribution of new PG material.** Bacteria were grown overnight in LB pH 4.6, the culture was then diluted 1:50 in LB pH 4.6 or pH 7.0 and further incubated at 37 °C for 2.5 h. Cells were centrifuged and resuspended in prewarmed LB pH 4.6 or pH 7.0 containing 3-[[[(7-Hydroxy-2-oxo-2*H*-1-benzopyran-3-yl)carbonyl]amino]-D-alanine hydrochloride (HADA). To localise the distribution of new peptidoglycan (PG) on the bacterial surface, cells were exposed to a pulse of 125 µM HADA for either 1 min or 30 min at 37 °C, and the flasks placed and vigorously shaken on ice with salt to immediately stop cell growth and prevent further HADA incorporation. Bacteria were washed three times with PBS, fixed with 3% PFA and processed for fluorescence microscopy as above.

**Protein extracts from intracellular bacteria.** NRK-49F fibroblasts were infected with *S.* Typhimurium strain MD5054 (*PBP2_SAL-3xFLAG*) using 500-$cm^2$ Nunclon Delta treated square BioAssay dishes (ThermoFisher) at a multiplicity of infection of 10:1 for 40 min. Bacteria were previously grown overnight in LB pH 5.8 at 37 °C in non-shaking conditions. Non-internalised bacteria were removed by three washes with prewarmed PBS, pH 7.4, supplemented with 0.9 mM $CaCl_2$ and 0.5 mM $MgCl_2$. The infected fibroblasts were incubated in fresh DMEM 10% FBS medium containing 100 µg/mL gentamicin up to 2 h post infection and the same medium with a lower amount of antibiotic, 10 µg/mL gentamicin, for the remaining period of infection.

At the required post-infection times, the fibroblasts were processed as described[71]. Briefly, infected cells were washed thrice in PBS, pH 7.4, and lysed with a scraper in 17 mL of lysis buffer (15% phenol, 19% ethanol, 0.4% SDS, 0.1 mg/ml DNase A) per 500-$cm^2$ cell culture plate. After 30 min at 4 °C, the lysate was collected in 40 mL polypropylene tubes (Sorvall) and centrifuged ($27,500 \times g$, 4 °C, 30 min). The resultant pellet enriched in intracellular bacteria was washed twice with 1 mL of a 1% basic phenol, 19% ethanol solution ($29,400 \times g$, 15 min, 4 °C) and, finally resuspended in 40 µL of Laemmli lysis buffer.

**In vivo protein cross-linking.** Overnight bacterial cultures were diluted 1:100 into 300 mL of LB pH 4.6. Bacteria were grown at 37 °C with shaking (150 rpm) until the culture reached an $OD_{600}$ of ~0.5. Bacteria were washed once with ice-cold PBS, pH 8.0 by centrifugation at $10,000 \times g$, 18 min, 4 °C. The pellets were resuspended in 1.6 mL of PBS pH 8.0 and split in two tubes. Disuccinimidyl suberate (DSS) or dithiobis[succinimidylpropionate] (DSP) ThermoScientific) were used as chemical cross-linkers and added at 2.5 mM final concentration. The reaction mixture was incubated at RT for 30 min and then quenched with 20 mM Tris (pH 7.5) for 15 min at RT. Bacteria were washed twice in 35 mL ice-cold Tris-buffered saline (TBS) buffer pH 7.5 and processed for preparation of membrane extracts.

**Immunoprecipitation of elongasome complexes.** Bacteria subjected to in vivo protein cross-linking with the DSS or DSP reagents were disrupted using a French Press (3 passes) in 15 mL of a buffer containing 50 mM Tris-HCl, pH 7.0, 60 µg/mL DNase A, and 2 mM phenyl-methylsulfonyl fluoride (PMSF) (Sigma-Aldrich). Unbroken cells were removed by centrifugation ($5000 \times g$, 5 min, 4 °C) and the supernatant subjected to ultra-centrifugation ($145,000 \times g$, 1 h, 4 °C). The pellet containing envelope material was resuspended with 400 µL of NP-40 buffer (150 mM NaCl, 1% NonidetP-40, 50 mM Tris pH 8.0) and incubated during 30 min at 4 °C with slow-rotation agitation. The sample was subjected to centrifugation ($25,000 \times g$, 1 h, 4 °C) and the supernatant containing inner membrane proteins and protein complexes used for immunoprecipitation with the respective antibodies. An amount corresponding to ~1–5 µg of immune-affinity-purified antibody (anti-PBP2) or commercial anti-tag antibodies (anti-FLAG, anti-HA) was added to these membrane lysates and the sample incubated at 4 °C with slow-rotation for 1.5 h.

To immunoprecipitate the elongasome complex, 1.5 mg (50 μL) of dynabeads protein G (Invitrogen) were incubated for 4 h with rotation at 4 °C with the sample containing antibody bound to complexes of cross-linked proteins. After this incubation, the sample was placed in a magnetic rack to remove the supernatant representing the flow-through. The magnetic beads to which the antibody-cross-linked protein complexes were bound, were further washed three times with the NP-40 buffer. A volume of 20 μL of Laemmli buffer with or without β-mercaptoethanol was then added to the samples cross-linked with either DSS or DSP, respectively, and boiled for 5 min before loading into acrylamide gels. To confirm the presence of large molecular weight elongasome complexes pulled down with the respective antibodies, samples were subjected to 10% SDS-PAGE and run until the dye front reached the bottom of the gel. The wells of the stacking gel were removed, and the gel equilibrated with transfer buffer and subsequently used for protein blotting to an Immobilon-P PVDF membrane in a wet transfer electrophoretic apparatus with a 200 mA current for 4 h. The apparatus was maintained at 4 °C during electrophoresis in a container with ice.

**Western blot analyses**. The following primary antibodies were used: mouse monoclonal anti-FLAG (Merck/Sigma-Aldrich, cat. no. F3165, dilution 1:5000), affinity-purified rabbit polyclonal anti-PBP2 (1:500; our lab collection) and mouse monoclonal anti-HA (BioLegend, catalogue no. 901533, dilution 1:2000). As secondary antibodies, goat polyclonal antibodies conjugated to horseradish peroxidase (HRP) anti-mouse-IgG (Bio-Rad cat. no. 1706516, dilution 1:20,000) and anti-rabbit IgG (Bio-Rad, cat. no. 1706515, dilution 1:30,000), were used. Protein transfer was made in hydrophobic polyvinylidene difluoride (PVDF) membranes with 0.45-μm pore size (Immobilon-P, Millipore, Sigma-Aldrich), previously soaked in methanol and subsequently equilibrated in transfer buffer composed of 48 mM Tris-HCl, 39 mM glycine, 0.036% SDS and 20% methanol, pH 8.5. Transfer was performed in semi-dry or wet transfer apparatuses, for either 1 h 15 V or 4 h 200 mA, respectively. After transfer, membranes were blocked in 5% (w/v) of skim milk in TBS-T (20 mM Tris-HCl pH 7.5, 150 mM NaCl, 0.1% Tween-20) for at least 1 h at RT. Incubations with the primary and secondary antibodies were made in the 5% skim milk-TBS-T solution for 18 h at 4 °C and 1 h at RT, respectively. Washes between incubations were performed for a minimum of three times with TBS-T buffer. Proteins recognised by the antibodies were detected with a Chemiluminescent Western ECL kit assay (Bio-Rad).

**Monitoring of morphological parameters in extracellular and intracellular bacteria**. Cellular dimensions (long and short axes) of individual cells in populations of extracellular bacteria were measured using the freely available software ObjectJ (https://sils.fnwi.uva.nl/bcb/objectj/)[69]. Since intracellular bacteria are commonly closely apposed in the phagosomal compartment of infected host cells, the automatic mode was avoided to prevent errors due to clustered bacteria being considered as a single object. Instead, dimensions of intracellular bacteria were measured manually with ObjectJ and the data collected with sample size of $n \geq 50$ individual cells. Cellular dimensions values of extra- and intracellular bacterial populations were further processed in GraphPrism version 9.5.0 to show dispersion and the average mean values. The measurements were performed in a minimum of two biological replicates.

**Proteomic analyses of elongasome complexes**. Once the presence of high-molecular-weight cross-linked elongasome complexes was verified by immunoblotting, the samples were subjected to 10% SDS-PAGE and the gel stained with Coomassie Blue. The

lanes containing blue-stained proteins were cut and divided into ten individual bands. Only bands spanning from the top of the lane, including stacking gel to the area corresponding to the monomeric form of the protein, were chosen for in-gel trypsin digestion using an Opentrons OT2 robot. The resulting tryptic peptides from each band were pooled to obtain a single sample per lane, except for the first set of samples named PROT1638_1, PROT1638_2, and PROT1638_3, corresponding to anti-FLAG-directed pull down of elongasomes in wild-type, $\Delta mrdA$ and $\Delta PBP2_{SAL}$ strains, which were analysed as sections A-B-C for each of them (see "Supplementary Data" for details). One hundred nanograms (100 ng) of each sample were analysed using a Thermo Ultimate 3000 liquid nanochromatograph coupled to an Orbitrap Exploris OE240 mass spectrometer in a Data-dependent acquisition format; the length of the gradient was 60 min. A reversed-phase C18 column (75 μm ID and 15-cm long) was used for separation. Both MS1 and MS2 spectra (~60,000–70,000 per sample) were used to launch a search against a database combining the reference proteome of S. Typhimurium reference strain SL1344 (entry UP000008962, Uniprot database), and some typical laboratory contaminants reaching a total number of 4775 entries. Peaks 7.5 was used as search engine, and the results were filtered at a false discovery rate (FDR) of 1% at the PSM level. The software Skyline v22.2 (https://skyline.ms/project/home/software/Skyline/begin.view) was used to extract the signal corresponding to protein-specific peptides identified in the different elongasome samples after pulling down with either anti-PBP2 or anti-FLAG (anti-PBP2$_{SAL-3xFLAG}$) antibodies and to illustrate in graphics their relative abundance.

**Phylogenetic analysis of PBP2$_{SAL}$ distribution in the Enterobacterales order**. A total of 325 bacterial genomes of the Enterobacterales order[72] were retrieved from NCBI (https://www.ncbi.nlm.nih.gov/genome/?term=enterobacterales/). Only complete genomes were selected with additional criteria as being catalogued as reference or representative genomes. These genomes were interrogated using the BLASTp tool using as query PBP2$_{SAL}$ from S. Typhimurium strain SL1344 (Uniprot entry E1WGF1). The assignment for presence/absence considered BLASTp scores with identities $\geq 60\%$ and coverage $\geq 90\%$ respect the PBP2$_{SAL}$ query. The sequences of these hits in fasta format were compiled. The phylogenetic tree was built using the compiled file with the BV-BRC GenTree tool with no end trimming or gappy sequences removal and using the "Randomised Axelerated Maximum Likelihood" (RAxML) algorithm with the LG general amino acid replacement matrix[73]. Trees were visualised using iTOL (https://itol.embl.de/) version 6.5.7.

**Statistics and reproducibility**. Experiments were performed in a minimum of two independent biological replicates. Confidence interval was established at 95%, inferring therefore statistical significance to differences between samples with a $P$ value $\leq 0.05$. In the analysis of elongation rates in bacteria at the onset of the stationary phase, GraphPrism version 9.5.0 was used to calculate by simple-linear regression the slope and standard deviation of length values corresponding to all individual cells measured in the 40–80 min period (40-, 60-, and 80-min points) for the distinct morphogenetic strains. The statistical significance of differences among the slopes was determined by the software Free Statistics Calculator, version 4.0, publicly available at https://www.danielsoper.com/statcalc/calculator.aspx?id=103.

## Data availability

A Supplementary-Data file is provided with the paper compiling source data behind all graphs and tables shown. The Supplementary Information file contains Supplementary Figs. 1–22 with their respective captions and Supplementary Table 1. Original uncropped

Western blots corresponding to Figs. 1c, 5a, 5b, 6a, 6b, 6c, 6d, Supplementary Fig. 1 and Supplementary Fig. 4 are shown in Supplementary Figs. 14, 15, 16, 17, 18, 19, 20, 21, and 22, respectively. The genome sequence of S. Typhimurium strain MD5052 (ΔmrdA) was uploaded to GenBank under BioProject ID PRJNA904137, Biosample accession SAMN31831537, SRA ID SRR22372222. The mass spectrometry proteomics data were deposited to the ProteomeXchange Consortium (https://www.proteomexchange.org) via the PRIDE partner repository (https://www.ebi.ac.uk/pride). These dataset receive the ProteomeXchange accession PXD039436 (http://www.ebi.ac.uk/pride/archive/projects/PXD039436) and are available for FTP download in ftp://ftp.pride.ebi.ac.uk/pride/data/archive/2023/08/PXD039436. The authors declare that the datasets generated or analysed in this study are included in this published article (and its Supplementary Information and Supplementary Data Files).

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

## Acknowledgements
We thank H. González for technical assistance; A. Paradela (Proteomics Unit, CNB-CSIC) for assistance with the proteomic data analysis and the Skyline software; J.C. Oliveros (Bioinformatics for Genomics and Proteomics Unit, CNB-CSIC) for assistance with the genome data of the ΔmrdA mutant; M.S. Van Nieuwenhze (Indiana University, USA) for the HADA fluorescent reagent; and, L.A. Fernández (CNB-CSIC) for the *E. coli* BW25141 strain. F.G.-d.P. was supported by grant PID2020-112971GB-I00/10.13039/501100011033 from the Spanish Ministry of Science and Innovation.

## Author contributions
S.C. and F.G.-d.P. designed research; S.C. performed research; S.C. and F.G.-d.P. analysed the data; F.G.-d.P. wrote the manuscript; the two authors revised and approved the submitted manuscript.

## Competing interests
The authors declare no competing interests.

## Inclusion and ethics statement
This study did not include any experimentation in animals or with biological material derived from humans.
