## [Peer Review File · Communications Biology]

Reviewers' comments:

Reviewer #1 (Remarks to the Author):

In this third iteration of their manuscript, the authors have done a better job of tailoring their conclusions to fit the data which is almost exclusively related to characterization of PBP2sal function. I am comfortable with their characterization of PBP2sal as an alternative class B PBP, which is capable of interacting with the elongasome independent of its cognate class B, PBP2. The presence of "extra" class B PBPs in other organisms suggests a role for these enzymes under specific conditions, which may or may not be host related, rendering the findings of interest to investigators interested in the bacterial cell envelope and cell envelope stress.

My only major comment is that the title, "A rod-shaped bacterium with two independent morphogenetic systems" is misleading. As the authors themselves demonstrate, PBP2 and PBP2sal interact interchangeably with the major components of the elongasome including the MreB complex on the cytoplasmic side and the transglycosylase, RodA, on the periplasmic side. Furthermore, the current title is confusing as rod shaped bacteria are already well known to contain two truly independent systems for cell wall synthesis: the divisome and the elongasome. Thus, to reflect their actual data and avoid confusing readers, the title and references to "independent elongasome" (e.g. line 401) should be updated to be more precise (e.g. Salmonella encodes two differentially regulated, functional copies of the class B PBP, PBP2)

Reviewer #2 (Remarks to the Author):

The paper "A rod-shaped bacterium with two independent morphogenetic systems" by Sónia Castanheria and Francisco García-del Portillo describes the study of how two class B PBPs, PBP2 and PBP2SAL, produced by *S. Typhimurium* in different growth conditions contribute to the development of a rod shape by participating in independent elongasome complexes with different elongation speeds.

Though the existence of alternative class B PBPs for elongation or division is known in other organisms (e.g. *Bacillus subtilis* produces 4 class B PBPs (bPBPs), two of them redundantly essential to achieve a rod shape [1]), this article demonstrate two important results: first, that the two bPBPs even though they assemble with mostly the same components they are part of independent complexes and, second, that the resulting complexes elongate the cell into rods at a different speed. In addition, the mass spec analysis of immunoprecipitated elongation complexes shows different preferences of the PBP2SAL and PBP2 elongasomes for carboxypeptidases and endopeptidases as components of these complexes.

Understandably, the investigation into the role of these proteins is at an early stage and, though the authors suggest a possible role for the alternative bPBP, PBP2SAL, in virulence (they speculate that the reduced elongation rate of the PBP2SAL elongasome, the preferred bPBP during intracellular growth, helps reduce damage to phagosomes and aid in achieving a low-growth persistence state), the evidence presented in this paper is not enough to resolve this issue and will need more experimental work.

This revised version addresses some of my previous concerns, but I still find some issues that could be improved.

Major issues

(1) I still find the results from Supporting Figure 5 and the interpretation by the authors (lines 154-174) confusing. In the revised version the authors have added statistical analysis of the results,

showing that the only statistically significant result from this assay is a reduction in cytosolic bacteria or damaged vacuoles in the $\Delta mrdA$ mutant 2h post-infection, which suggests that PBP2SAL, the only active elongation bBPB is this mutant, contributes to maintaining the integrity of the phagosome. However, it is counterintuitive in my opinion that the WT cells, which the authors demonstrated previously that it produces almost exclusively PBP2SAL in this condition, should behave any different than the $\Delta mrdA$ mutant. And because of the lack of statistical significance we cannot compare with the conditions in which PBP2 is the only elongation bBPB. Thus I think this assay does not help to clarify PBP2SAL role in virulence, it is confusing and probably not necessary in the paper.

(2) lines 294-295 "...suboptimal configuration of aPBPs and bPBPs exploited by *S. Typhimurium* to lower the growth rate": This statement makes no sense. What is a suboptimal configuration of aPBPs and bPBPs and how can that lower the growth rate? Are they implying that the aPBP that becomes part of an elongosome can affect the growth rate? The authors do not provide enough evidence to support this claim with just the minor effect on the final OD in the growth curves of strains with or without deletions of *mrcA* or *mrcB* in a PBP2 or PBP2SAL background (Supp fig. 13). Moreover, the growth rate (slope of the linear part of the curve) is not measured in that figure. In my opinion the only conclusion that can be taken from the results in Supp Figs. 10-13 is that aPBPs do not affect the shape of the cells in the different conditions measured, so presumably both aPBPs in *Salmonella* can perform their role in the elongosome complexes.

Minor issues/Comments

(1) In the discussion section, pages 431-441, the authors discuss the different preference for DD-CPase and endopeptidases (EPases) of the PBP2 and PBP2SAL elongosomes. They still state that the balance between TPase and EPase/CPase rates "could ultimately define elongation rate" (line 436). I think the authors forget that the elongation rate (defined as rate of incorporation of PG to the side wall in an ordered way), also depends on the glycosyltransferase rates (by RodA, which forms a complex with either PBP2 and PBP2SAL). I am not sure what the contribution of CPases is to elongation rates, though arguably EPase activity is required to incorporate new material to existing PG and probably the fact that PBP2 and PBP2SAL have different CPases and EPases is more to do with the fact that different hydrolases might be specially in activity at different pHs, like it has been observed in other organisms. If the authors want to speculate about why different bPBPs produce different PG incorporation rates, they should include RodA in the discussion. Perhaps a structural comparison of the PBP2-RodA and PBP2SAL-RodA complexes would be more informative (see recent papers describing the structure of *E. coli* PBP2-RodA complex).

(2) lines 251-252: the authors cite copy numbers for MreB in *Bacillus subtilis*. They can also cite the copy numbers in *E. coli* ranging from 2400-11300 molecules/cell depending on the growth condition [2].

(3) Improve contrast of plate Supp Fig 7c PCN medium pH 4.6

(4) Supp Fig. 12. The colours in the legends do not always match the colours in the plots. I also suggest showing line plots instead of symbols for these growth curves, to improve clarity.

(5) I think in some parts of the manuscript the authors should check whether they are using the correct term to refer to related but different and easy-to-confuse terms such as growth rate (increase in cell mass over time), cell expansion (increase in cell volume), cell wall growth (incorporation of new material into the peptidoglycan sacculus) or cell envelope expansion (increase in cell surface, which does not necessarily involve incorporation of new material to PG as turgor can make the sacculus stretch more or less). The relationship between cell mass growth and cell wall and cell envelope expansion has been explored previously in the literature (check for example recent papers from the Van Teeffelen group).

[1] Wei et al. 2003 J Bacteriol 185:4717-4726

[2] Li et al. 2014 Cell 157:624-35 and <https://ecocyc.org/ECOLI/substring-search?type=NIL&object=mreB>

Responses to reviewers of manuscript #COMMSBIO-23-2187-T

Reviewer #1 (Remarks to the Author):

In this third iteration of their manuscript, the authors have done a better job of tailoring their conclusions to fit the data which is almost exclusively related to characterization of PBP2sal function. I am comfortable with their characterization of PBP2sal as an alternative class B PBP, which is capable of interacting with the elongasome independent of its cognate class B, PBP2. The presence of “extra” class B PBPs in other organisms suggests a role for these enzymes under specific conditions, which may or may not be host related, rendering the findings of interest to investigators interested in the bacterial cell envelope and cell envelope stress.

My only major comment is that the title, “A rod-shaped bacterium with two independent morphogenetic systems” is misleading. As the authors themselves demonstrate, PBP2 and PBP2sal interact interchangeably with the major components of the elongasome including the MreB complex on the cytoplasmic side and the transglycosylase, RodA, on the periplasmic side. Furthermore, the current title is confusing as rod shaped bacteria are already well known to contain two truly independent systems for cell wall synthesis: the divisome and the elongasome. Thus, to reflect their actual data and avoid confusing readers, the title and references to “independent elongasome” (e.g., line 401) should be updated to be more precise (e.g., *Salmonella* encodes two differentially regulated, functional copies of the class B PBP, PBP2)

Response: We thank the reviewer for the overall positive comment to our work and for indicating that our results are of interest for investigators in the fields of bacterial morphogenesis and cell envelope stress.

With regards to the change in the title, we only partially agree with that proposed by the reviewer. In the original title we introduced the statement “...two independent morphogenetic systems...” precisely to avoid confusion given the known presence of two systems, one promoting cell elongation and the other cell division (elongasome, divisome), that alternate during the cell cycle and are therefore dependent one of the other. The word “independent” was thought to transmit to the reader the new concept that bacteria like *Salmonella* use two morphogenetic systems in two different conditions or niches, depending on the regulatory signals perceived. On the other hand, we agree with the reviewer that as such it may be misleading regarding the main message of the study, the existence of two differentially regulated elongasome complexes.

In our opinion, the title proposed by the reviewer does not reflect, however, the novelty of the study. We have already reported in previous studies that two functional copies of bPBP consisting of PBP2 and PBP2_{SAL} exist in *Salmonella* and that both are differentially regulated by signals as acidic pH, osmolarity and amount of nutrients in the medium (Castanheira et al. *EBioMedicine*, 2020, PMID: 32344200; López-Escarpa et al., *Molecular Microbiology*, 2022, PMID: 36115022). Based on these previous publications, we therefore proposed for this manuscript a new title as:

“Evidence of two differentially regulated elongasomes in *Salmonella*”

This title highlights the novelty of two elongasomes that are present in this bacterium at the time that no confusion exists with the divisome. Please, note that Reviewer #2 indicates in his/her comments that the identification of two independent elongasomes in *Salmonella* is the most important result of our work and, as such, we believe it must be credited in the title.

Reviewer #2 (Remarks to the Author):

The paper “A rod-shaped bacterium with two independent morphogenetic systems” by Sónia Castanheira and Francisco García-del Portillo describes the study of how two class B PBPs, PBP2 and

PBP2SAL, produced by *S. Typhimurium* in different growth conditions contribute to the development of a rod shape by participating in independent elongosome complexes with different elongation speeds.

Though the existence of alternative class B PBPs for elongation or division is known in other organisms (e.g. *Bacillus subtilis* produces 4 class B PBPs (bPBPs), two of them redundantly essential to achieve a rod shape [1]), this article demonstrate two important results: first, that the two bPBPs even though they assemble with mostly the same components they are part of independent complexes and, second, that the resulting complexes elongate the cell into rods at a different speed. In addition, the mass spec analysis of immunoprecipitated elongation complexes shows different preferences of the PBP2SAL and PBP2 elongasomes for carboxypeptidases and endopeptidases as components of these complexes.

Response: We thank the reviewer for the positive comments to our work and for highlighting the two most important results of our work: the existence of two bPBPs that assemble in independent complexes and the distinct elongation speed shown by those complexes.

We are also now citing in the revised version the work in *Bacillus subtilis* by Wei et al. (*J. Bacteriol.* 2003, PMID: 12896990) indicating the apparent redundancy between PPBP2a and PBP2 for cell elongation during vegetative growth and sporulation (lines 331-332). We consider this phenomenon different to the one we describe in *Salmonella*. In this latter case, PBP2 and PBP2_{SAL} promote cell elongation responding to distinct environmental cues.

Understandably, the investigation into the role of these proteins is at an early stage and, though the authors suggest a possible role for the alternative bPBP, PBP2SAL, in virulence (they speculate that the reduced elongation rate of the PBP2SAL elongosome, the preferred bPBP during intracellular growth, helps reduce damage to phagosomes and aid in achieving a low-growth persistence state), the evidence presented in this paper is not enough to resolve this issue and will need more experimental work.

This revised version addresses some of my previous concerns, but I still find some issues that could be improved.

Major issues

(1) I still find the results from Supporting Figure 5 and the interpretation by the authors (lines 154-174) confusing. In the revised version the authors have added statistical analysis of the results, showing that the only statistically significant result from this assay is a reduction in cytosolic bacteria or damaged vacuoles in the Δ mrda mutant 2h post-infection, which suggests that PBP2SAL, the only active elongation bPBP is this mutant, contributes to maintaining the integrity of the phagosome. However, it is counterintuitive in my opinion that the WT cells, which the authors demonstrated previously that it produces almost exclusively PBP2SAL in this condition, should behave any different than the Δ mrda mutant. And because of the lack of statistical significance, we cannot compare with the conditions in which PBP2 is the only elongation bPBP. Thus, I think this assay does not help to clarify PBP2SAL role in virulence, it is confusing and probably not necessary in the paper.

Response: We agree with the reviewer in the preliminary stage of these observations. However, please note that PBP2 is still present in wild type bacteria at the post-infection times examined (see supplementary Figure S4). This result was previously discussed in the responses to reviewers corresponding to previous versions of this manuscript. The presence of “both” bPBPs (PBP2, PBP2_{SAL}) in intracellular bacteria when using the tissue culture model hampered a clear interpretation concerning the role played by each bPBP inside the host cell. This could however be solved in part by the single mutants lacking one or the other bPBP. Importantly, the *in vivo* data that we published in 2020 demonstrated that is PBP2_{SAL}, and not PBP2, the preferred bPBP produced by *Salmonella* inside the host (Castanheria et al. *EBioMedicine*, 2020, PMID: 32344200). This apparent discrepancy between the *in vitro* and the *in vivo* models regarding the relative amount of both bPBPs reflects the

limitation of tissue cultures compared to animal models. We are nonetheless currently planning experiments in other host cell types like macrophages, aiming to see differences regarding the stability of the phagosomal membrane when either the PBP2 or the PBP2_{SAL}-elongasome operate.

Since we agree with the reviewer in the still preliminary stage of these observations obtained in the *in vitro* infection model, we have followed his/her recommendation for keeping this supplementary figure S5 for future work focused on the role of these two bPBPs in *Salmonella* virulence. Therefore, we are not showing these data in the current manuscript.

Please, note that in the responses to previous revised versions, we incorporate some unpublished information supporting a cross-talk between the activity of PBP2 or PBP2_{SAL} and that of the two type-III secretion systems encoded by the pathogenicity islands SPI-1 and SPI-2. We are therefore much confident of these two bPBPs involved in cell elongation contributing to *Salmonella* virulence.

(2) lines 294-295 "...suboptimal configuration of aPBPs and bPBPs exploited by *S. Typhimurium* to lower the growth rate": This statement makes no sense. What is a suboptimal configuration of aPBPs and bPBPs and how can that lower the growth rate? Are they implying that the aPBP that becomes part of an elongasome can affect the growth rate? The authors do not provide enough evidence to support this claim with just the minor effect on the final OD in the growth curves of strains with or without deletions of *mrcA* or *mrcB* in a PBP2 or PBP2_{SAL} background (Supp fig. 13). Moreover, the growth rate (slope of the linear part of the curve) is not measured in that figure. In my opinion the only conclusion that can be taken from the results in Supp Figs. 10-13 is that aPBPs do not affect the shape of the cells in the different conditions measured, so presumably both aPBPs in *Salmonella* can perform their role in the elongasome complexes.

Response: We thank the reviewer for this comment related to a statement that it may be seen as ambiguous. Our intention was to speculate on the possibility of a distinct configuration of aPBPs and bPBPs (in terms of stoichiometry and relative abundance) that could differ from that employed by extracellular bacteria in laboratory media. We agree the study does not provide support for this assumption, so we have deleted the corresponding sentence limiting the conclusion to that commented by the reviewer considering the data shown in supplementary figures 10-13. Thus, we affirm in the revised version that both aPBPs can contribute to insert PG material working in coordination with any of the two elongasomes identified (**lines 409-411**).

Minor issues/Comments

(1) In the discussion section, pages 431-441, the authors discuss the different preference for DD-CPase and endopeptidases (EPases) of the PBP2 and PBP2_{SAL} elongasomes. They still state that the balance between TPase and EPase/CPase rates "could ultimately define elongation rate" (line 436). I think the authors forget that the elongation rate (defined as rate of incorporation of PG to the side wall in an ordered way), also depends on the glycosyltransferase rates (by RodA, which forms a complex with either PBP2 or PBP2_{SAL}). I am not sure what the contribution of CPases is to elongation rates, though arguably EPase activity is required to incorporate new material to existing PG and probably the fact that PBP2 and PBP2_{SAL} have different CPases and EPases is more to do with the fact that different hydrolases might be specially in activity at different pHs, like it has been observed in other organisms. If the authors want to speculate about why different bPBPs produce different PG incorporation rates, they should include RodA in the discussion. Perhaps a structural comparison of the PBP2-RodA and PBP2_{SAL}-RodA complexes would be more informative (see recent papers describing the structure of *E. coli* PBP2-RodA complex).

Response: We agree with the reviewer and apologize for not including the important bPBP-RodA interaction in the discussion as a probable factor behind the distinct elongation rates assigned to the two elongasomes identified. In the new revised version, we included this possibility (**see lines 346-358**).

We also cite in the revised version the very recent paper of Shlosman et al. in Nature Communications 2023 (PMID: 37301887) (lines 349-351). This study demonstrates two conformational states (open and close) in the PBP2-RodA complex that undergoes dynamic changes, being the open complex the active conformation. The authors conclude that such dynamics may influence the rate at which cell elongation occurs. We discuss in the revised version of our manuscript that these data have been obtained entirely in neutral pH and that it will be of interest for future studies to test activity of the complex in both neutral and acidic pH, especially when assessing the activity of the PBP2_{SAL}-RodA complex.

We have also analysed *in silico* complexes of PBP2-RodA and PBP2_{SAL}-RodA predicted with the AlphaFold software that do not show major differences (see below, Figure 1). In our view, this however does not mean that subtle differences in the distance of interacting residues in the interfaces I and II as defined by Sjodt et al. (Nature Microbiology PMID: 32152588), may occur. We are now intensively searching for these putative differences to perform site-directed mutagenesis and infer whether the changes affect cell elongation rate in different genetic backgrounds. We also consider the fact that AlphaFold does not integrate in the prediction parameters as pH, which may alter considerably the PBP2_{SAL} folding. Our long-term objective is to isolate the PBP2_{SAL}-RodA complex and perform *in vitro* polymerization (transglycosylation) assays at different pH values examining in parallel the canonical PBP2-RodA complex.

Figure 1. RodA-PBP2 and RodA-PBP2_{SAL} interactions as predicted by AlphaFold. In both cases, the two proteins were artificially fused as single polypeptide before assessing structure prediction. Note the similarity in the interface in which the two proteins of each pair interact.

(2) lines 251-252: the authors cite copy numbers for MreB in *Bacillus subtilis*. They can also cite the copy numbers in *E. coli* ranging from 2400-11300 molecules/cell depending on the growth condition [2].

Response: As proposed, the study of Li et al. in 2004 is now included in the new revised version citing the number of MreB molecules estimated to be present in *E. coli* (lines 232-236)

(3) Improve contrast of plate Supp Fig 7c PCN medium pH 4.6

Response: As requested, contrast has been improved in this figure for this specific plate. Please, note that this figure has been renumbered as Supplementary figure 6 after removing previous Supplementary figure 5 following recommendation of this reviewer.

(4) Supp Fig. 12. The colours in the legends do not always match the colours in the plots. I also suggest showing line plots instead of symbols for these growth curves, to improve clarity.

Response: We have mounted in the revised version new supplementary figures S11 and S12 using line plots with lower number of symbols to improve clarity. Matching of the colours has been reviewed and corrected.

(5) I think in some parts of the manuscript the authors should check whether they are using the correct term to refer to related but different and easy-to-confuse terms such as growth rate (increase in cell mass over time), cell expansion (increase in cell volume), cell wall growth (incorporation of new material into the peptidoglycan sacculus) or cell envelope expansion (increase in cell surface, which does not necessarily involve incorporation of new material to PG as turgor can make the sacculus stretch more or less). The relationship between cell mass growth and cell wall and cell envelope expansion has been explored previously in the literature (check for example recent papers from the Van Teeffelen group).

Response: We thank very much the reviewer for this comment and have reviewed these terms and correct them accordingly throughout the manuscript.

[1] Wei et al. 2003 J Bacteriol 185:4717-4726

[2] Li et al. 2014 Cell 157:624-35 and <https://ecocyc.org/ECOLI/substring-search?type=NIL&object=mreB>

REVIEWERS' COMMENTS:

Reviewer #2 (Remarks to the Author):

This latest revision of the paper "A rod-shaped bacterium with two independent morphogenetic systems" by Sónia Castanheria and Francisco García-del Portillo has now addressed all my previous concerns and I did not find any outstanding major or minor issues.

The results described in this manuscript will be of great interest to the community of researchers interested in bacterial morphogenesis and focuses on an alternative cell elongation complex that it is used by *S. Typhimurium* during infection, so it will be important for researchers studying virulence in *Salmonella* and other intracellular bacterial pathogens.